# *Culex* Mosquito Piwi4 Is Antiviral against Two Negative-Sense RNA Viruses

**DOI:** 10.3390/v14122758

**Published:** 2022-12-10

**Authors:** Elizabeth Walsh, Tran Zen B. Torres, Claudia Rückert

**Affiliations:** Department of Biochemistry and Molecular Biology, College of Agriculture, Biotechnology & Natural Resources, University of Nevada, Reno, NV 89557, USA

**Keywords:** mosquito, *Culex*, Piwi4, vpiRNA, orthobunyavirus, La Crosse orthobunyavirus, RNAi

## Abstract

*Culex* spp. mosquitoes transmit several pathogens concerning public health, including West Nile virus and Saint Louis encephalitis virus. Understanding the antiviral immune system of *Culex* spp. mosquitoes is important for reducing the transmission of these viruses. Mosquitoes rely on RNA interference (RNAi) to control viral replication. While the siRNA pathway in mosquitoes is heavily studied, less is known about the piRNA pathway. The piRNA pathway in mosquitoes has recently been connected to mosquito antiviral immunity. In *Aedes aegypti*, Piwi4 has been implicated in antiviral responses. The antiviral role of the piRNA pathway in *Culex* spp. mosquitoes is understudied compared to *Ae. aegypti*. Here, we aimed to identify the role of PIWI genes and piRNAs in *Culex quinquefasciatus* and *Culex tarsalis* cells during virus infection. We examined the effect of PIWI gene silencing on virus replication of two arboviruses and three insect-specific viruses in *Cx. quinquefasciatus* derived cells (Hsu) and *Cx. tarsalis* derived (CT) cells. We show that Piwi4 is antiviral against the La Crosse orthobunyavirus (LACV) in Hsu and CT cells, and the insect-specific rhabdovirus Merida virus (MERDV) in Hsu cells. None of the silenced PIWI genes impacted replication of the two flaviviruses Usutu virus (USUV) and Calbertado virus, or the phasivirus Phasi-Charoen-like virus. We further used small RNA sequencing to determine that LACV-derived piRNAs, but not USUV-derived piRNAs were generated in Hsu cells and that PIWI gene silencing resulted in a small reduction in vpiRNAs. Finally, we determined that LACV-derived DNA was produced in Hsu cells during infection, but whether this viral DNA is required for vpiRNA production remains unclear. Overall, we expanded our knowledge on the piRNA pathway and how it relates to the antiviral response in *Culex* spp mosquitoes.

## 1. Introduction

*Culex* species mosquitoes are vectors for several viruses that infect humans, birds, horses, and other vertebrates. *Culex* spp. mosquitoes are globally distributed and are a significant global health and economic concern. Despite their global prevalence and threat to human, equine, and avian species, they remain understudied compared to other mosquito vectors, such as *Aedes aegypti* or *Anopheles gambiae*. Worldwide, *Culex* spp. mosquitoes transmit a variety of viruses, including West Nile virus (WNV) [1], Usutu virus (USUV) [2], and Japanese encephalitis virus (JEV) [3]. Infection with these arthropod-borne viruses (arboviruses) is usually nonpathogenic to mosquitoes, and arboviruses persist in mosquitoes throughout their lifespan [4]. However, many arboviruses cause disease in humans and other vertebrates. Vaccines are often unavailable, increasing the need for effective vector control and other strategies to block arbovirus transmission [5]. Studying the numerous factors contributing to viral infection and dissemination in the mosquito will contribute to developing new approaches for reducing arbovirus transmission by mosquitoes [6].

In mosquitoes, the RNA interference (RNAi) pathway is one of the major innate immune pathways that control arbovirus infection [7]. RNAi is conserved across nearly all eukaryotes and results in the silencing of gene expression through the repression or degradation of complementary RNA. There are three distinct pathways mediated by small-interfering RNAs (siRNAs), micro RNAs (miRNA), and P-element-induced wimpy testis (PIWI)-interacting RNAs (piRNAs) [7]. The siRNA pathway plays an important role in controlling virus replication in most virus–vector combinations that have been studied [8]. However, more recently, the piRNA pathway has been connected to mosquito immune responses to arbovirus infection and is becoming an increasingly important area of research [9]. In most eukaryotes, PIWI proteins and piRNAs are typically only expressed in germline tissue and are involved in silencing transposable elements [10]. The biogenesis of endogenous piRNAs consists of the primary piRNA pathway and a secondary amplification pathway. In the *Drosophila* primary pathway, piRNA precursors are transcribed by RNA polymerase II from genomic piRNA clusters [11] and cleaved by the endonuclease Zucchini (Zuc) into single-stranded piRNA intermediates [12]. After 5′ end generation, piRNA intermediates are loaded into Piwi and Aubergine (Aub) [13], and 3′ end trimming occurs as the final step to mature piRNA generation [14,15]. Primary piRNAs are usually antisense to transposons and have a 5′ U_1_ bias [16]. While Piwi shows steady-state localization to the nucleus in *Drosophila* germ line cells [17], Aub and Argonaute-3 (Ago3) localize to the nuage of germline cells and are involved in the secondary amplification pathway. Secondary piRNAs are a result of target cleavage by Aub. Primary piRNAs will guide Aub to a complementary target RNA molecule and Aub will cleave exactly 10 nucleotides upstream from the primary piRNA 5′ end, resulting in secondary piRNAs that have a 5′ A_10_ bias [18,19]. These secondary piRNAs associate with Ago3 and will mediate cleavage of target RNA (e.g., precursor piRNAs or negative-sense transposon RNA), producing piRNAs similar to primary piRNAs [20]. Mosquitoes share common piRNA pathway features with *Drosophila* [21], such as a U_1_/A_10_ bias from ping-pong amplification which is observed in both virus-derived piRNAs (vpiRNAs) [22,23] and transposable element-derived piRNAs (TE-piRNAs) [24]. Additionally, while *Aedes* spp. and *Culex* spp. have 7 PIWI proteins [25,26], there is no direct ortholog to *Drosophila* Aub or Piwi proteins, so the exact mechanisms of piRNA biogenesis in mosquitoes are still unclear. De novo production of vpiRNAs has been reported in several mosquito species [27]; however, not all virus infections result in vpiRNA production [28,29]. In *Ae. aegypti*, vpiRNAs are produced from a variety of arboviruses, including chikungunya virus (CHIKV), dengue virus (DENV), Sindbis virus (SINV), Semliki Forest virus (SFV), and Rift Valley Fever virus (RVFV) [22,30,31,32,33,34,35]. In *Cx. quinquefasciatus,* production of vpiRNAs derived from the insect-specific rhabdovirus Merida virus (MERDV) [26,29], Culex densovirus [26], and the phlebovirus RVFV [36] has been observed. In *Culex tarsalis,* vpiRNAs have only been observed from the insect-specific phasivirus Phasi Charoen-like virus (PCLV) [26,28]. It is unknown how vpiRNAs are produced, although it might require the production of a viral cDNA form (vDNA) [37,38]. In *Aedes albopictus* U4.4 cells, inhibition of CHIKV vDNA reduced vpiRNAs and increased CHIKV susceptibility, but this response was less prevalent in vivo and also reduced siRNA production [38]. Similarly, inhibiting SINV vDNA in *Ae. aegypti* Aag2 cells led to increased SINV replication and reduced vpiRNAs [37]; however, it remains unclear whether vpiRNAs directly affect viral replication and transmission in vivo.

In *Cx. quinquefasciatus* and *Ae. aegypti* mosquitoes, *piwi* genes have expanded [25,39], resulting in seven *piwi* genes in each species. However, not all are directly orthologous to each other between the two species. Four *piwi* genes are expressed in *Ae. aegypti* Aag2 cells (Piwi4, Piwi5, Piwi6, Ago3) [27,40], but only *Ae. aegypti* Piwi4 (AePiwi4) has been implicated as an antiviral effector protein [37,41]. Knockdown of AePiwi4 increased replication of several different arboviruses [42,43,44], but there are contradictory reports on the involvement of AePiwi4 in piRNA production. On the one hand, knockdown of AePiwi4 increased SFV replication in Aag2 cells but did not reduce vpiRNAs [34], while knockdown of AePiwi4 during SINV infection decreased mature vpiRNAs in a separate study [37]. Piwi4 involvement in vpiRNA production could be virus-dependent, or interpretation may vary depending on the research methodology and data analysis. Others have further shown that vpiRNA production does require the Tudor protein AeVenemo, which recruits AeYB [45], AePiwi5, and AeAgo3 and forms a protein complex that facilitates piRNA production [33,46]. AePiwi5 is also necessary for TE-derived piRNAs [33]. AePiwi6 and AePiwi4 appear involved in, but not essential, for some TE-derived piRNAs [33], and AePiwi6 is involved in the production of DENV-derived vpiRNAs [32]. Neither AePiwi5 nor AeAgo3 have been implicated in antiviral response, and knockdown of these proteins did not affect SINV [33], supporting the hypothesis that vpiRNA production may, in fact, not be required for the antiviral activity of AePiwi4. AePiwi4 may be involved in or bridge several pathways, as it associates with proteins involved in both the siRNA pathway and the piRNA pathway, as well as small RNAs themselves [41,47,48]. Additionally, AePiwi4 was found to bind sense vpiRNAs produced during SINV infection and EVE-derived piRNAs [41], but the impact of this on virus replication is unclear. *Ae. aegypti* aBRAVO, a protein that mediates antiviral activity against positive-strand RNA arboviruses, has also been found to interact with AeDcr2 and AePiwi4 but did not affect the exo-siRNA pathway when knocked down, indicating a complex network of interactions between RNAi pathways [49]. Less is known about the piRNA pathway and its potential antiviral role in *Culex* compared to *Aedes* species. In *Cx. quinquefasciatus*, CqPiwi4, 5, and 6, and piRNA pathway proteins Zuc and Ago3 are expressed in the midgut, while all PIWI proteins are expressed in ovaries [29]. It has not yet been determined if any *Culex* PIWI proteins have an antiviral role or which proteins contribute to piRNA biogenesis in *Culex* spp. mosquitoes. Additionally, it is still largely unknown for either *Culex* spp. or *Aedes* spp. if vpiRNAs contribute to any observed antiviral response, or if antiviral PIWI proteins act independently of vpiRNAs.

Here, we characterized the impact of PIWI gene silencing on the replication of arboviruses and insect-specific viruses in both *Cx. quinquefasciatus* (Hsu) cells and *Cx. tarsalis* (CT) cells. We used an orthobunyavirus (La Crosse encephalitis virus, LACV) and a flavivirus (Usutu virus, USUV) as two arboviruses from different families, which replicate in these cells. For insect-specific viruses, we investigated a rhabdovirus (Merida virus; MERDV) in Hsu cells, while in CT cells we investigated a flavivirus (Calbertado virus, CLBOV), and a phasivirus (Phasi-Charoen-like virus, PCLV), which is a member of the *Bunyavirales*. The insect-specific viruses persistently replicate within these cell lines. Following transfection of gene specific dsRNA to transiently reduce the expression of PIWI genes (as well as Ago3 and the endonuclease Zuc), we determined the impact on replication of LACV, USUV, MERDV (Hsu), CLBOV (CT), and PCLV (CT). We also examined the effect of Piwi4 overexpression in Hsu cells on LACV replication and analyzed the small RNA (sRNA) response in Hsu cells to MERDV, LACV, and USUV. We then determined whether transiently silencing select PIWI genes impacts small RNA composition in Hsu cells. Lastly, we investigated if vDNA forms are generated in response to LACV infection in Hsu cells. Overall, we found that Piwi4 is antiviral in *Cx. quinquefasciatus* and *Cx. tarsalis* cells, and silencing PIWI genes resulted in small decreases in vpiRNAs of select lengths.

## 2. Materials and Methods

### 2.1. Cell Lines

The *Cx. quinquefasciatus* ovary-derived (Hsu) cell line [50] was grown at 27 °C, and 5% CO_2_ in Dulbecco’s Modified Eagle Medium (DMEM; Corning #10-013-CV; Corning, New York, NY, USA) supplemented with 10% FBS and antibiotics (100 units/mL penicillin, 100 μg/mL streptomycin, 5 μg/mL gentamicin). The *Cx. tarsalis* embryo-derived (CT) cell line [51] was grown at 27 °C, and ambient air in Schneider’s Drosophila medium (Genesee Scientific #25-515, San Diego, CA, USA) supplemented with 7% FBS and antibiotics (100 units/mL penicillin, 100 μg/mL streptomycin, 5 μg/mL gentamicin). The *Ae. aegypti* embryo-derived (Aag2) cell line [52] was grown at 27 °C and ambient air in Schneider’s Drosophila medium supplemented with 7% FBS and antibiotics (100 units/mL penicillin, 100 μg/mL streptomycin, 5 μg/mL gentamicin).

### 2.2. Viruses

LACV strain R97841d was kindly provided by Brandy Russell, CDC Fort Collins, as a Vero p1 stock. This isolate was initially obtained from a human brain in Tennessee in 2012. We propagated the virus on Vero cells and concentrated virus using the Amicon^®^ Ultra-15 Centrifugal Filter (Sigma-Aldrich, #UFC9010 St. Louis, MO, USA) to obtain a high titer stock. Virus stock titers were determined by standard plaque assay on Vero cells.

USUV strain TMN was kindly provided by Dr. James Weger-Lucarelli (Virginia Tech) as a Vero p1 stock. It was originally isolated in the Netherlands in 2016 from a common blackbird (*Turdus merula*). We propagated the virus on Vero cells and concentrated the virus using the Amicon^®^ Ultra-15 Centrifugal Filter (Sigma-Aldrich, #UFC9010 St. Louis, MO, USA) to obtain a high titer stock. Virus stock titers were determined by standard plaque assay on Vero cells.

The insect-specific rhabdovirus MERDV was previously identified as a persistent infection in Hsu cells [53]. No stocks were grown, but MERDV RNA levels in cell culture were quantified in various experiments. Similarly, the insect-specific flavivirus Calbertado virus (CLBOV) persistently infects CT cells [29]. No stocks were grown, but RNA levels of persistent virus were quantified in various experiments. Finally, the insect-specific phasivirus Phasi-Chareon-like phasivirus (PCLV) persistently infects CT cells [28]. No stocks were grown, but RNA levels of persistent virus were quantified in various experiments.

### 2.3. Phylogenetic Analysis

*Cx. quinquefasciatus*, *Ae. aegpyti*, and *Drosophila melanogaster* PIWI, Zuc, and Ago3 protein sequences were collected from the VectorBase and the NCBI database. *Culex tarsalis* coding sequences were identified from the published *Cx. tarsalis* genome [54] using Blast command line. Both Blastn and Blastp were used to identify *Cx. tarsalis* PIWI proteins with E values of < 10^−40^. Additionally, a BlastP search was done using the PAZ/PIWI domain to determine any additional PIWI proteins. A PAZ/Piwi domain was verified to be present in all identified *Cx. tarsalis* PIWI proteins. MEGAX was used for phylogenetic analysis of full-length PIWI family gene sequences. Sequences were aligned with the MUSCLE algorithm with the UPGMA cluster method. Phylogenetic relationships among proteins were estimated using a maximum likelihood analysis of the amino acids and the LG model. A bootstrap analysis was done using 500 pseudoreplicates.

### 2.4. Generation of dsRNA

Gene-specific dsRNA was generated using the MEGAScript™ RNAi kit (Thermo Fisher, #AM1626, Waltman, MA, USA) according to the manufacturer’s instructions. Briefly, PCR primers containing the T7 promoter sequence were designed for *Cx. quinquefasciatus* and *Cx. tarsalis* PIWI genes as well as *zuc* and *ago3* (Table 1). Hsu and CT cell cDNA generated using the High-Capacity cDNA reverse transcription kit (Thermo Fisher, #4368814, Waltman, MA, USA) was used as a template to amplify target regions for dsRNA synthesis. As a non-specific control dsRNA, GFP dsRNA was generated using a GFP containing plasmid and T7 primers (Table 1). Target regions were amplified by PCR using Platinum™ SuperFi II PCR Master Mix (Thermo Fisher, #12368010 Waltman, MA, USA) and verified with gel electrophoresis. PCR products were purified and concentrated using Zymo DNA-25 clean and concentrator kit (Zymo Research, #D4033; Irvine, CA, USA). DsRNA was generated via in vitro transcription using the MEGAScript™ RNAi kit (Thermo Fisher, AM1626, Waltman, MA, USA). Concentrations and purity of dsRNA were determined by Thermo Scientific™ NanoDrop™ One spectrophotometer and gel electrophoresis.

### 2.5. Transfection of dsRNA

Hsu or CT cells were seeded into 24-well plates in 1 mL of culture medium. Cells were incubated at 27 °C overnight to allow for cell adhesion. Transfection of gene-specific dsRNA, a non-specific dsRNA control (GFP), and a no dsRNA control was performed using Lipofectamine RNAiMAX (Thermo Fisher, #13778075, Waltman, MA, USA) following the manufacturer’s protocol. Briefly, per well, 1.5 µL Lipofectamine RNAiMAX reagent (Thermo Fisher, #13778100, Waltman, MA, USA) was diluted in 25 µL Opti-MEM (Thermo Fisher, #31985062, Waltman, MA, USA). In parallel, 500 ng dsRNA were diluted in 25 µL Opti-MEM (Thermo Fisher, #31985062, Waltman, MA, USA). Next, the diluted dsRNA was added to the diluted Lipofectamine RNAiMAX reagent (Thermo Fisher, AM1626, Waltman, MA, USA), mixed by pipetting up and down, and the complex was incubated for 5 min at room temperature. An amount of 50 µL of the complex was added to cells with 450 µL fresh complete media. Cells were incubated for 48 h before virus infection. For MERDV samples, plates were incubated 96 h before RNA extraction.

### 2.6. Virus Infection

Hsu or CT cells were seeded into 24-well plates and infected with USUV at an MOI of 50 or LACV MOI 10. These were MOIs previously established in the lab as required to infect 70–90% of the cells. Briefly, per well, virus was diluted in 250 µL of the respective culture media (DMEM or Schneider’s without additives) and added to the well after removal of complete culture media; for uninfected controls, 250 µL media was added to the well. Cells were incubated for 2 h at 27 °C to allow for infection. After 2 h, the virus-containing media was removed, and replaced with 1 mL of complete culture media. The plate was then incubated at 27 °C for 48 h.

### 2.7. RNA Extraction

Following the manufacturer’s instructions, RNA was extracted using the Directzol RNA miniprep kit (Zymo Research, #D4033; Irvine, CA, USA). Briefly, culture media was removed from all wells, and 300 µL Trizol was added to each well. Plates were incubated with TRIzol^®^ on a rocker for 20 min to ensure complete lysis. Cell lysates were transferred to a microcentrifuge tube and frozen at −80 °C prior to RNA extraction following the provided protocol. RNA was quantified using a Qubit Flex Fluorometer (Thermo Fisher, Waltham, MA, USA).

### 2.8. Quantitative Real-Time PCR

cDNA was first generated from 100 ng of each RNA sample using the High-Capacity cDNA reverse transcription kit (Thermo Fisher, #4368814, Waltman, MA, USA). Quantitative real-time PCR (qRT-PCR) was then performed using iTaq Universal SYBR Green Supermix (Bio-Rad, #1725120, Hercules, CA, USA) or iTaq Universal Probes 1 step kit (Bio-Rad, #1725141, Hercules, CA, USA) on a CFX96 Touch Real-Time PCR Detection System (Bio-Rad). All RT-qPCR primers are listed in Table 2. Each target gene or viral RNA was normalized to a previously tested, reliable housekeeping gene (Hsu: actin 5; CT: actin 1) to quantify either gene silencing levels or virus replication.

For the Piwi4 overexpression experiment, LACV RNA was quantified using an RNA standard curve and normalized input RNA levels, as opposed to normalization to actin 5.

### 2.9. Generation of Piwi4 Overexpression Plasmid

The Ac5-STABLE1-neo plasmid from Addgene (plasmid #32425; referred to as pAc_GFP here) was used as the backbone of the plasmids discussed in this study. To produce the pUb_eGFP plasmid, the *Drosophila* promoter of the Ac5-STABLE1-neo plasmid was replaced with an *Ae. aegypti* polyubiquitin promoter cloned from Addgene plasmid #162161. The CqPiwi4 overexpression construct (pUb_Piwi4) was then constructed by replacing the GFP sequence with the CqPiwi4 coding region tagged with an N-terminal 3xFLAG sequence for antibody recognition. The Piwi4 coding region was obtained by extracting total RNA from Hsu cells using Direct-zol RNA Miniprep (Zymo Research, #D4033; Irvine, CA, USA), generating cDNA using the High-Capacity RNA-to-cDNA™ Kit (Applied Biosystems, #4387406, Waltham, MA, USA), and amplifying the entire cds. The plasmid backbone, the N-terminal 3xFLAG-tag sequence from Addgene plasmid #49330, and the target gene of interest were amplified via PCR with Q5^®^ Hot Start High-Fidelity Master Mix (New England Biolabs, # M0515, Ipswich, MA, USA). The PCR products were isolated using a Zymoclean Gel DNA Recovery kit (Zymo Research, # D4001, Irvine, CA, USA) and subsequently combined into a single plasmid using Gibson Assembly^®^ Cloning (New England Biolabs, #E5510S, Ipswich, MA, USA). The assembled plasmid was transformed into High Efficiency NEB Stable Competent E. coli (New England Biolabs, #C3040H, Ipswich, MA, USA) for plasmid amplification. Plasmids were isolated using ZymoPURE II Plasmid Maxiprep Kit (Zymo Research, #NC1121573; Irvine, CA, USA). The sequences of the finalized constructs were verified via restriction digestion and Sanger sequencing.

### 2.10. Transfection of CqPiwi4 Expression Plasmid

Hsu cells were seeded in a 24-well plate and incubated at 27 °C and 5% CO_2_ overnight. At the day of transfection, the complete cell media was replaced with 200 µL of Opti-MEM (Thermo Fisher Scientific, #31985062, Waltham, MA, USA) and placed back in the incubator while the lipoplex was prepared. The cells were then transfected with pUb_Piwi4 or pUb_EGFP plasmids using X-tremeGENE™ HP DNA transfection reagent (Roche, #6366236001, Burlington, MA, USA) per the manufacturer’s protocol. Briefly, plasmid DNA concentration was normalized to 100 ng/μL. Per well, 500 ng plasmid (5 µL) was combined with 44 μL of Opti-MEM™ I Reduced Serum Medium (Thermo Fisher Scientific, #31985062, Waltham, MA, USA) prior to the addition of 1 μL of X-tremeGENE™ HP DNA transfection reagent (Roche, #6366236001, Burlington, MA, USA). After 20 min of incubation, the diluted transfection reagent complex was added to the cells. After 1 h of incubation, 1 mL of complete cell medium was added to the cells. Mock-transfected cells were treated with X-treme GENE Transfection reagent diluted in Opti-MEM alone. All experiments were performed in triplicates. The expression of CqPiwi4 was confirmed through immunostaining using a monoclonal ANTI-FLAG M2 antibody (Sigma Aldrich #F3165, Burlington, MA, USA) at two days post-transfection.

### 2.11. Immunostaining

Cells were fixed with 4% paraformaldehyde for 20 min. Paraformaldehyde was removed, and cells were washed with 1x PBS 3 times for 5 min each. Cells were incubated in a blocking buffer (1x PBS/5% Normal Goat Serum/0.3% Triton™ X-100) for 1 h. A 1:200 dilution of the primary antibody (anti-FLAG M2 antibody, Sigma Aldrich #F3165, Burlington, MA, USA) in an antibody dilution buffer (1x PBS/1% BSA/0.3% Triton™ X-100) was added to cells and incubated at 4 °C overnight. Cells were then rinsed with 1x PBS 3 times for 5 min each, and a 1:1000 dilution of an Alexa Fluor^®^ 488 conjugated anti-mouse IgG (Cell Signaling, #4409S, Danvers, MA, USA) in an antibody dilution buffer (1x PBS/1% BSA/0.3% Triton™ X-100) was added to the cells. This secondary antibody was incubated for 1 h at room temperature in the dark, and cells were then washed with 1x PBS 3 times for 5 min each. Cells were counterstained with a 1:10,000 dilution of DAPI and incubated at room temperature for 10 min in the dark. Cells were washed once with 1x PBS. Cells in 100 µL of 1x PBS were visualized using a Keyence BZ-X710.

### 2.12. Small RNA Library Preparation and Sequencing

Small RNA sequencing libraries were prepared using the NEBNext Small RNA Library Prep Set for Illumina (New England Biolabs, #E7330L, Ipswich, MA, USA), following the manufacturer’s protocol. The 3′ adaptor was ligated for 18 h at 16 °C as recommended by the manufacturer to enrich for piRNAs. Final Libraries of 140–160 bps were gel purified for accurate size selection using the Zymoclean Gel DNA recovery kit (Zymo Research, #D4001, Irvine, CA, USA). Libraries were quantified by the NEBNext Library Quant Kit for Illumina (New England Biolabs, #E7630L, Ipswich, MA, USA) and pooled at equimolar ratios to a concentration of 15 nM. The pooled library concentration was verified using both the NEBNext Library Quant Kit for Illumina^®^ (New England Biolabs, #E7630L, Ipswich, MA, USA) and a Qubit Flex Fluorometer (Thermo Fisher). Agilent TapeStation was used to confirm the accurate size distribution of the library (140–160 bp). Due to a small peak at ~95 bp, the library pool was further purified using an SDS-PAGE purification prior to sequencing (Nevada Genomics Center). The final pooled library was sequenced at the Nevada Genomics Center using 1 × 50 bp reads on an Illumina NextSeq2000 P3 kit (Illumina, #20046810, San Diego, CA, USA).

### 2.13. Sequencing Analysis

All sRNA sequencing data were analyzed using a previously established and described pipeline [29,56] with only minor modifications. Briefly, FASTQ files were trimmed off the 3′ adapter using FASTX Toolkit (http://hannonlab.cshl.edu/fastx_toolkit/; accessed for installation on 9 March 2022), size selected (19–32 nt reads, 26–28 nt reads, or 29–30 nt reads) and aligned to the viral consensus sequences using Bowtie 1.3.1 [57] allowing for 1-mismatch. Bowtie-generated SAM output files were used as the input for processing through SAMtools [58]. Histograms were generated from the individual read counts to show the overall size and polarity distribution of virus-derived sRNA reads. The mpileup function of SAMtools was used to determine nucleotide targeting of the viral genome by sRNA reads. Nucleotide bias at specific sRNA read positions was analyzed and plotted (seqLogo and motifStack) using the R package viRome [59]. We also used viRome to analyze 5′ read distance (read.dist.plot) to determine if a 10 nt read overlap, indicative of ping-pong amplification, was present in our suspected vpiRNA reads.

### 2.14. vDNA Detection

Hsu or Aag2 cells were seeded into 24-well plates in 1 mL of culture medium. Cells were incubated overnight at 27 °C to allow for cell adhesion. Cells were infected with LACV at MOI 10 as described above. RNA or DNA was extracted at either 2 days post-infection (dpi) or 6 dpi using the Directzol RNA miniprep kit (Zymo Research, #D4033; Irvine, CA, USA) or the Quick-DNA miniprep kit (Zymo Research, #D3025; Irvine, CA, USA), respectively. Primers to amplify approximately 200 base pair long regions (Table 3) were designed to cover the S segment of the LACV genome. Actin cDNA and DNA was detected as a positive control. RNA and DNA were normalized to 40ng/µL. cDNA was made from RNA samples using the High-Capacity cDNA reverse transcription kit (Thermo Fisher, #4368814, Waltman, MA, USA). Q5^®^ Hot Start High-Fidelity 2X Master Mix (New England Biolabs, # M0515, Ipswich, MA, USA) was used for all PCRs to verify the presence of viral cDNA, actin cDNA, and actin DNA, and to determine the presence of vDNA in infected samples. Gels were imaged using a Bio-Rad Gel Doc XR+ Imaging system.

## 3. Results

### 3.1. Phylogenetic Analysis of Piwi Genes in Culex spp. Mosquitoes

We first wanted to identify and determine the relationship between *Cx. tarsalis* PIWI genes and *Cx. quinquefasciatus* PIWI genes. *Cx. tarsalis* is closely related to *Cx. quinquefasciatus* and is an important vector of WNV in large parts of the United States [1,60]. The *Cx. tarsalis* genome was assembled in 2021 [54]. We first identified four PIWI genes (Piwi 1, 4, 5, 6) present in the *Cx. tarsalis* genome through a BLAST search with homologous *Cx. quinquefasciatus* protein sequences (Appendix A). To verify that all PIWI genes were found, we also searched for PAZ/PIWI domains which returned two additional genes, Piwi7 and Piwi8, not detected in the original search. All identified genes had PAZ/PIWI domains. To determine the phylogeny of *Cx. tarsalis* and *Cx. quinquefasciatus* Piwi proteins in relation to *Ae. aegypti* and *D. melanogaster,* we generated a maximum likelihood phylogenetic tree (Figure 1). *Cx. tarsalis* PIWI names were chosen based on homology to *Cx. quinquefasciatus* and *Ae. aegypti* PIWI proteins. While six PIWI genes were identified in *Cx. tarsalis*, only 5 (Piwi1, Piwi4, Piwi5, Piwi6, Piwi7) were expressed in *Cx. tarsalis*-derived CT cells. The coding sequences of Piwi7 and Piwi8 were highly similar, differing only by a few nucleotides. We thus hypothesize that Piwi8 may be a sequencing artifact and not a true gene encoded by the *Cx. tarsalis* genome. Additionally, homologs to Ago3 and Zuc were identified, and both were expressed in CT cells.

### 3.2. Cx. quinquefasciatus and Cx. tarsalis Piwi4 Are Antiviral In Vitro

We first wanted to characterize the effect of PIWI proteins on virus replication in *Cx. quinquefasciatus*-derived Hsu cells. Using an RNAi-based method, we silenced Piwi1-7 (separately and together), Zuc, and Ago3 expression in Hsu cells with gene-specific long dsRNA. Long dsRNA targeting GFP was used as a negative control. It is important to note that Piwi1 and Piwi3 are highly homologous, and siRNAs produced from Piwi1 or Piwi3 derived dsRNA may target both genes. We first determined the impact of PIWI gene silencing on the insect-specific rhabdovirus MERDV (Figure 2a,b), from which vpiRNAs were previously detected in Hsu cells [29]. DsRNA transfection resulted in a significant ≥50% decrease in the expression of all genes, with the exception of Piwi6b in samples where all PIWI genes were simultaneously silenced (Figure 2a). Silencing Piwi4 expression resulted in a significant (*p* < 0.001) 3-fold-increase in cellular MERDV RNA compared to the GFP control (Figure 2b). Silencing of other PIWI genes (e.g., Piwi1) also mildly increased cellular MERDV RNA, but this was not significant across experiments. There was also a significant (*p* < 0.01) 1.8-fold increase in cellular MERDV RNA when Piwi1-7 were silenced together (Figure 2b).

Next, we investigated the effect of PIWI gene silencing on two arboviruses: the orthobunyavirus LACV (Figure 2c,d) and the flavivirus USUV (Figure 2e,f). In LACV-infected samples, expression was significantly reduced for all PIWI genes except Piwi7 when silenced individually and for all genes when silenced together (Figure 2c). Silencing of Piwi4 resulted in a significant (*p* < 0.0001) 3-fold increase in LACV cellular RNA (Figure 2d). There was also a significant (*p* < 0.01) 1.9-fold increase in LACV cellular RNA when Piwi1-7 were silenced together (Figure 2d). In USUV-infected samples, expression of all targeted genes was ≥50% reduced, but only significantly (*p* < 0.05) for Piwi1, Piwi3, and Piwi4 in individually silenced samples and Piwi1 when silenced together; this is likely due to the variable PIWI gene expression in the control samples (Figure 2e). While none of the gene silencing conditions significantly increased cellular USUV RNA (Figure 2f), there was a trend for an increase when Piwi5 was silenced. We thus looked at the effect of Piwi5 silencing at further time points post-USUV infection. Gene expression was significantly reduced by at least 50% at all time points (Figure 2g). Although there was an observed increase in USUV cellular RNA at 7 dpi, it was not significant (Figure 2h).

To determine if *Cx. tarsalis* PIWI genes are involved in antiviral immune responses, we silenced the expression of the newly identified PIWI genes (Figure 1) in *Cx. tarsalis* CT cells (Figure 3). Piwi1-7, Zuc, and Ago3 expression was silenced in *Cx. tarsalis* CT cells using sequence-specific dsRNA, and dsRNA targeting GFP was used as a control.

We looked at two insect-specific viruses that persistently infect CT cells PCLV, a phasivirus within the order *Bunyavirales,* and CLBOV, a flavivirus. Both viruses were quantified in the same PIWI gene-silenced samples. A significant >50% decrease in expression was detected for all genes (Figure 3a). When Piwi1-7, Zuc, and Ago3 expression was silenced in CT cells, there was no significant effect on PCLV (Figure 3b) or CBLOV (Figure 3c) cellular RNA 4 days post transfection.

We next investigated the effect of PIWI gene silencing on LACV (Figure 3d,e) and USUV (Figure 3f,g) replication in CT cells. Gene expression was significantly reduced by ≥50% for all PIWI genes in LACV-infected samples (Figure 3d). Silencing Piwi4 resulted in a significant 2-fold increase (*p* < 0.001) in LACV cellular RNA (Figure 3e). In USUV-infected samples, expression was significantly reduced by ≥50% for all PIWI genes (Figure 3f), but gene silencing did not significantly increase USUV cellular RNA (Figure 3g). Overall, silencing Piwi4 expression increased MERDV (Hsu) and LACV (Hsu, CT) cellular RNA.

### 3.3. Overexpression of CqPiwi4 Reduces LACV Replication

Given that silencing Piwi4 gene expression increased LACV and MERDV RNA in Hsu cells, we wanted to determine the effect of Piwi4 overexpression on virus replication. We transfected Hsu cells with a Piwi4-FLAG expression plasmid or a control GFP expressing plasmid and determined LACV and MERDV replication. Initially, we detected Piwi4 expression by immunostaining for the 3x-FLAG tag (Figure 4a) which confirmed that cells were transfected and expressed Piwi4-FLAG. Piwi4 overexpression was further validated at 1 and 2 days post-transfection using RT-qPCR to measure Piwi4 mRNA abundance. In samples transfected with the overexpression plasmid, there was a significant 100-fold and 109-fold increase in Piwi4 expression at 1 and 2 days post-transfection, respectively, compared to the GFP control (Figure 4b). Piwi4 overexpression significantly decreased LACV RNA levels at both 1 dpi and 2 dpi compared to the GFP control (Figure 4c).

### 3.4. vpiRNAs Are Produced during LACV and MERDV Infection

It is currently unknown if Piwi4 is involved in the piRNA biogenesis pathway and if piRNAs are involved in an immune response. We used sRNA sequencing to determine the sRNA response to virus infection in Hsu cells. We first characterized if virus-derived sRNAs were observed for MERDV, LACV, and USUV in untreated Hsu cells, with MERDV serving as a positive control due to previously detected MERDV-derived piRNAs in Hsu cells [29]. We prepared triplicate samples of untreated cells infected with LACV MOI 10, USUV MOI 50, or left uninfected. Cellular RNA for all samples was extracted at 2 dpi, sRNA libraries were prepared and sequenced using Illumina NextSeq2000. MERDV mapping reads showed a clear peak of 21 nt reads, indicative of siRNAs, and a substantial peak from 26 to 28 nt, indicating the presence of piRNAs (Figure 5a). The 21 nt reads mapped to the MERDV sense and antisense RNA (Figure 5a), while 26–28 nt reads mapped predominantly to the sense RNA strand. Additionally, a strong nucleotide bias was seen for a uridine at position 1 (shown as ‘T’ in sequenced cDNA libraries) and an adenine at position 10 of these 26–28 nt reads, indicative of the ping-pong amplification cycle that occurs during piRNA biogenesis (Figure 5b).

In LACV-infected cells, we analyzed sRNA reads by segment (Figure 5c–h). The majority of sRNA reads mapped to the LACV S segment, where we observed a strong peak at 21 nt targeting sense and antisense strands (Figure 5c). We also observed peaks at 26–28 nt on the sense strand and a peak at 29–30 nt on the antisense strand (Figure 5c). Interestingly, we observed another clear peak of positive sRNA reads that mapped to the LACV S segment (Figure 5c). The vast majority of these 19 nt reads were derived from one sequence found in the 3′ UTR region of the sense strand. We confirmed that this sRNA read sequence (ATGGGTGGGTGGTAGGGGA) was not host-derived—it did not map the *Cx. quinquefasciatus* genome and it was not present in uninfected samples. When analyzing nucleotide bias of 26–28 nt reads derived from the LACV S segment, a strong U_1_ and A_10_ bias was observed (Figure 5d) and 5′ read distance analysis showed a clear enrichment for a 10 nt overlap (Appendix A). Given that for sRNAs derived from the LACV S segment, there was a peak at 26–28 nt corresponding to positive sRNA reads and a peak at 29–30 nt reads corresponding to negative sRNA reads, we also looked at the nucleotide bias from 29 to 30 nt. A strong U_1_ and A_10_ bias was also observed for those reads (Appendix A) and a clear 10 nt overlap was observed in the 5′ read distance plot (Appendix A), indicating that these slightly larger sRNAs also represent vpiRNAs.

Fewer sRNA reads mapped to the LACV M segment (Figure 5e), but there was still a distinct peak at 21 nt and when analyzing the less abundant 26–28 nt reads, there was a strong U_1_ and slight A_10_ bias, indicative of some vpiRNA production at lower levels (Figure 5f). However, analysis of the 5′ read distance (Appendix A) showed no enrichment for a 10 nt overlap, possibly indicating a lack of ping-pong amplification.

Even less sRNA reads were derived from the L segment (Figure 5g). There was a peak at 21 nt, but among the longer 24–32 nt reads, only a small peak at 26–28 nt on the antisense strand was observed (Figure 5g). When nucleotide bias for these reads was analyzed, there was no clear bias on the positive-sense reads, but a strong bias for a U_1_ on the antisense reads (Figure 5h), indicative of the generation of primary piRNAs without ping-pong amplification. A lack of ping-pong amplification was supported by the 5′ read distance analysis, which showed no enrichment for a 10 nt overlap (Appendix A).

In USUV-infected cells, USUV-derived sRNAs displayed a strong peak at 21 nt both for the sense and antisense strand but reads trailed off at the larger sizes with no distinct peak (Figure 5i). We used the few detected 26–28 nt USUV-derived reads to determine nucleotide bias, but no U_1_ or A_10_ nucleotide bias was seen, indicating that USUV-derived vpiRNAs are not produced in Hsu cells (Figure 5j).

### 3.5. Piwi Silencing Affects sRNA Populations in Cx. quinquefasciatus Cells

We next wanted to determine if any (and which) of the PIWI genes in *Cx. quinquefasciatus* are involved in vpiRNA production for MERDV, USUV, LACV, or general piRNA production. Since we based this experiment at the time on the originally identified PIWI genes by Campbell et al. [25], we did not include Piwi7 here. We prepared triplicate sRNA library samples from Hsu cells in which we had silenced expression of Piwi1-6b and Zuc using long dsRNA. Additionally, we prepared triplicate sRNA library samples from Piwi4 silenced Hsu cells infected with LACV and Piwi5 silenced Hsu cells infected with USUV, based on their antiviral effects on these viruses (trend for USUV). Cells treated with GFP dsRNA were used as a control for all samples. The numbers of total 19–32 nt sRNA sequencing reads and those that mapped to MERDV, LACV (all segments), and USUV genome are shown in Table 4. While the percentage of viral reads was low overall, the observed percentages for MERDV were comparable to previous reports [29] and viral sRNA reads for USUV and LACV were within the expected range for 2 dpi. Interestingly, infection with USUV or LACV increased MERDV sRNA reads, possibly suggesting a complex interaction of these viruses with the mosquito RNAi system.

We first looked at the effect of PIWI gene silencing on sRNA composition in general. We compared the total 19–32 nt sRNA population from Piwi1-6b dsRNA treated, uninfected (but persistently MERDV infected) Hsu cells. We observed most significant differences at 21 nt reads (Figure 6a), with an increase in 21 nt reads after silencing of Piwi1, Piwi4, Piwi5, Piwi6a, Piwi6b, and Zuc (Figure 6a). Piwi5 silencing resulted in the highest proportion of siRNA reads per million total reads (Figure 6a). In contrast, 22 and 23 nt reads were generally decreased in PIWI-silenced samples to varying degrees (Figure 6a). We next focused on larger 24–32 nt reads as a representative for potential piRNAs, which were less abundant overall (note change in y-axes for Figure 6b,c). We found that Piwi5 silencing resulted in significantly fewer reads compared to the GFP control for 24 and 26 nt reads (Figure 6b). All other PIWI genes appeared to also reduce piRNA biogenesis or stability to some degree, but these trends were not significant, except for the 24 nt reads. No significant differences were observed in 29–32 nt reads (Figure 6c).

Since we noticed a clear trend for reduced piRNA counts in the PIWI gene-silenced samples, we summed up counts from 24 to 32 nt reads to determine if the overall reduction of these putative piRNAs was significant (Figure 6d). Piwi1, Piwi3, and Zuc silencing showed a trend in 24–32 nt read reduction that was not significant, but silencing of Piwi4, Piwi5, Piwi6a, and Piwi6b significantly reduced the overall portion of 24–32 nt reads, explaining the proportional increase in 21 nt reads. The biggest impacts were seen for Piwi5 (67% reduction) and Piwi 6b (56% reduction).

### 3.6. Piwi Silencing Affects Insect-Specific Virus sRNA Populations in Cx. quinquefasciatus Cells

We then compared the effect of PIWI gene silencing on MERDV-derived sRNA reads; sRNA reads that mapped to the MERDV genome were normalized to the total amount of viral sRNA reads to focus on proportional changes in sRNA size classes. We noticed an increase in positive and negative-sense 21 nt siRNAs for all PIWI-silenced samples (Figure 7a). For larger size classes (Figure 7b,c), we observed a reduction in sRNAs after PIWI gene silencing. Specifically, positive-sense 26–28 nt reads were reduced to varying degrees after Piwi1, Piwi3, Piwi4 and Piwi6b silencing, with the most consistent decrease for Piwi4 and Piwi6b (Figure 7b). For the negative-sense reads, we saw the most significant decreases in 25 and 29 nt reads after PIWI gene silencing (Figure 7b,c), with the most impact observed after Piwi5, Piwi6a, and Piwi6b silencing. Zuc silencing had no significant impact on any specific MERDV-derived sRNA size class except for a mild increase at the 21 nt read length. While the impacts on individual size classes of potential MERDV-derived piRNAs were relatively small, we also summarized all 24–32 nt reads (Figure 7d) and determined the overall impact of PIWI gene silencing on these putative piRNAs. We found that positive-sense reads were significantly reduced after silencing of Piwi1, Piwi3, Piwi4, Piwi6b, matching what we saw on individual sizes. Furthermore, we found that silencing Piwi5, Piwi6a, and Piwi6b resulted in the biggest reduction in negative-sense 24–32nt MERDV-derived reads. We also detected small reductions in these negative-sense reads after Piwi1, Piwi4, and Zuc silencing compared to the GFP control; however, these small reductions are difficult to interpret because we also observed a small reduction in 24–32 negative-sense reads in our NTC control.

Overall, we concluded that Piwi1, Piwi3, and Piwi4 predominantly reduced positive piRNAs, Piwi5 and Piwi6a reduced negative-sense piRNA, while Piwi6b appeared to significantly reduce both positive and negative-sense reads.

Additionally, we looked at the effect of PIWI gene silencing on the positional mapping MERDV-derived sRNAs (Appendix A). We did not observe any differences in the specific regions where 19–23 nt (Appendix A) or 26–28 nt (Appendix A) sRNAs mapped between conditions. However, we did observe that MERDV-derived siRNAs primarily mapped to a specific region in the L gene and MERDV-derived piRNAs mapped primarily to the N gene as previously described [29].

### 3.7. Piwi Silencing Affects Arbovirus-Derived sRNA Populations in Cx. quinquefasciatus-Derived Cells

Since LACV RNA only increased when Piwi4 was silenced (Figure 2d), we chose to sequence LACV-infected samples where Piwi4 was silenced, as well as a GFP and NTC control. We focused our analysis on sRNAs that mapped to the LACV S segment, given the much higher proportion of reads compared to the L and M segments (Figure 5). Additionally, we chose to sequence USUV-infected samples where Piwi5 was silenced, given the (nonsignificant) trend for an increase observed in USUV RNA after Piwi5 silencing (Figure 2f).

For LACV, Piwi4 silencing resulted in significantly more (*p* < 0.001) positive and negative LACV-derived 21 nt siRNA reads compared to the GFP control (Figure 8a). However, there were significantly fewer LACV-derived negative-sense 29 nt (*p* < 0.01) and 30 nt (*p* < 0.001) reads in Piwi4 silenced samples (Figure 8a). We also noted a small (non-significant) reduction in positive-sense LACV-derived reads of 26 and 27 nt length in Piwi4 silenced cells compared to the GFP control (Figure 8a). When we took the sum of all 24–32 nt reads and compared Piwi4 silenced samples to the GFP control, we saw a significant reduction in negative-sense reads after Piwi4 silencing (Figure 8b).

Additionally, we looked at the effect of PIWI gene silencing on the positional mapping of LACV (S segment) derived piRNAs. We did not observe any differences in the specific regions where 19–23 nt or 26–28 nt sRNAs mapped between conditions (Appendix A).

In USUV-infected cells, Piwi5 silencing resulted in significantly more (*p* < 0.001) USUV-derived 21 nt siRNA reads compared to the GFP control (Figure 8c). In contrast, we observed a significant decrease in USUV-derived 22 nt sRNAs. Indeed, 22 nt is a miRNA size class and it is unclear what role 22 nt virus-derived sRNAs play during virus infection and how (and if) these may be linked to Piwi5. As expected, very few sRNA reads mapped to the USUV genome at 26–28 nt for any sample, further supporting that USUV-derived piRNAs are not produced in *Cx. quinquefasciatus* derived Hsu cells at a significant level (Figure 8c). We did sum up all counts of 24–32 nt USUV-mapping reads (Figure 8d) and while there was no significant difference in 24–32 nt sRNA reads after Piwi5 silencing, we observed a trend for less sRNAs in this class. It is important to note that these read counts are derived from less than 1000 total reads per sample of this size group, making any strong conclusions difficult.

### 3.8. CqPiwi4 Silencing Has Only Minor Impacts on TapiR1 Abundance

It has recently been suggested that Piwi4 antiviral activity could be mediated through its role in gene regulation via the endogenous piRNAs tapiR1 and tapiR2 [61,62]. Since we had performed PIWI gene silencing and small RNA sequencing, we mined our data for reads aligning to tapiR1 (CTCTTCAAAACTAGGTCGTTTTAGAATATT) and tapiR2 (TTTCGGATATGTTTTAGAAATTCGTTTTT) to determine if CqPiwi4 silencing resulted in a reduction of tapiR1 and/or tapiR2 which may explain the observed antiviral effects. We identified a large amount of reads identical to the full-length 30 nt tapiR1 (Figure 9a) and 29 nt tapiR2 (Figure 9b) sequences, as well as reads that were 3 nt shorter, i.e., 27 and 26 nt reads, respectively. For tapiR1, reads with 20 nt length were also abundant. While we identified reads of almost all 19–30 nt lengths that matched to tapiR1 and tapiR2, we visualized only those read lengths with high abundance and significant differences between PIWI gene silencing conditions. Expression of tapiR2 was generally higher in all samples compared to tapiR1.

When comparing PIWI gene-silenced samples to the GFP control, we noticed that while most PIWI genes had some impact on tapiR1 abundance, this was only significant for Piwi5 at 20 nt reads, Piwi3 and Piwi5 at 27 nt reads, and Piwi3, Piwi4, Piwi5, and Piwi6b at the full length tapiR1 read length of 30 nt. Piwi3 silencing resulted in the most dramatic decrease in tapiR1 abundance. Piwi3 silencing also significantly decreased full length (29 nt) tapiR2 abundance. The 26 nt reads mapping to tapiR2 were significantly reduced when Piwi1 and Piwi5 were silenced. Interestingly, Zuc and Piwi6a silencing resulted in a small, but significant, increase in tapiR2 abundance. While PIWI gene silencing clearly impacted tapiR1 and tapiR2 abundance, there was no clear correlation between abundance of tapiR1, tapiR2 and CqPiwi4 specifically. It is still possible that CqPiwi4 is bound to tapiR1 and tapiR2, but our data suggest that expression and stability/binding of these piRNAs may also be mediated through other PIWI genes in Hsu cells, such as Piwi3 and Piwi5.

### 3.9. LACV Viral DNA Forms Are Produced in Hsu Cells during Infection

While we previously could not detect vDNA from MERDV in Hsu cells [29], evidence suggests that vpiRNA production may require vDNA in *Ae. aegypti* Aag2 cells [37]. We thus wanted to determine if LACV vDNA was produced following infection of Hsu cells with LACV. Most LACV-derived piRNAs mapped to the coding region of the LACV S segment (Figure 5c); therefore, we designed overlapping primers spanning the entire coding region of the LACV S segment to detect vDNA forms in Hsu and Aag2 cells (Figure 10a). LACV vDNA is known to be produced in infected Aag2 cells [63], serving as a suitable positive control for our experiment. We infected both Aag2 and Hsu cells with LACV and extracted genomic DNA 2 dpi and 6 dpi since vDNA might not be detectable during early infection periods. Additionally, we included cDNA from LACV-infected cells as a positive control of infection and validation for PCR conditions. Actin was also included as a positive control to ensure DNA was present in samples (Appendix A). We found that vDNA was detectable in Aag2 and Hsu cells at both 2 dpi and 6 dpi only with primer set 7, spanning from 696 to 909 bp in the LACV S genome segment (Figure 10b,c). In uninfected Aag2 and Hsu cells, we verified that no band was detected using this primer set to rule out non-specific binding or the presence of an EVE with a related sequence (Appendix A). When comparing where along the LACV S segment vDNA is produced to where piRNAs mapped (26–30 nt), we observed an overlap at the strongest positive-sense peak (Figure 10d). However, vDNA was not detected from other regions with strong vpiRNA coverage along the LACV S segment (Figure 10d). Additionally, only few piRNAs map to the 3′ end of the coding region of the LACV S segment, where vDNA was detected (Figure 10d). A strong peak of 19–23 nt reads did fall into the genome region where vDNA was detected (Figure 10e), potentially supporting the role of vDNA as a booster of sRNA responses in general.

## 4. Discussion

In the present study, we characterized how different PIWI genes in *Culex* spp. cells affect viral replication and what effect PIWI gene silencing has on sRNA composition and virus-derived sRNA responses in *Cx. quinquefasciatus* Hsu cells. Overall, we found that Piwi4 is antiviral against the arbovirus LACV in *Cx. quinquefasciatus* and *Cx. tarsalis* cells, and the insect-specific virus MERDV in *Cx. quinquefasciatus* cells. Additionally, we found that LACV, USUV, and MERDV generated a strong exogenous siRNA response in *Cx. quinquefasciatus* cells. However, vpiRNAs were only produced in response to LACV and MERDV infection. MERDV-derived piRNAs were previously reported in Hsu cells [29]. While LACV-derived piRNAs had not been reported in *Cx. quinquefasciatus* cells or mosquitoes, vpiRNAs derived from other members of the *Bunyavirales* have been reported in *Culex* spp., including Rift Valley fever virus [36] and PCLV [28]. Lastly, we determined that PIWI gene silencing, including Piwi4, affected both the total sRNA and virus-derived sRNA read composition. Together, our data indicate that Piwi4 in *Culex* spp. is antiviral and likely involved in vpiRNA biogenesis or processing. We also show that all PIWI genes have some impact on piRNA biogenesis and/or stability, with Piwi5 and Piwi6b having the biggest impact on overall cellular piRNA production.

Piwi4 has previously been identified as antiviral against many arboviruses and insect-specific viruses in *Aedes* spp. [37,41]. We show that Piwi4 is also antiviral in *Cx. quinquefasciatus* and *Cx. tarsalis* cells against one arbovirus, LACV, and one insect-specific virus, MERDV. Piwi4 overexpression in Hsu cells also decreased LACV RNA, confirming our gene silencing results. Our research supports that the antiviral function of Piwi4 is conserved between *Culex* spp. and *Aedes* spp. mosquitoes. However, Piwi4 was not antiviral against USUV, PCLV, and CBLOV, suggesting that *Culex* Piwi4 may be less broadly active. We initially expected Piwi4 to be antiviral against PCLV, given that Piwi4 is antiviral against the two other negative-sense single-stranded RNA viruses we assessed and previous reports that PCLV-derived piRNAs are generated in CT cells [28]. It is possible that complete knockout or longer exposure to dsRNA may have had an impact on PCLV replication. We did not observe any impact of Piwi4 silencing on USUV or CLBOV replication. Since we also did not detect any USUV-derived piRNAs here and no CBLOV-derived piRNAs were detected in our previous study [29], it is possible that the effect of Piwi4 is vpiRNA-dependent, but that presence of vpiRNAs is not the only factor (see PCLV). Overall, our data support that the antiviral effect of Piwi4 in *Culex* spp. is virus dependent and that the role of vpiRNAs themselves remains unclear.

Most of the LACV-derived piRNA reads in Hsu cells mapped to the S segment of the virus, similar to previous reports for PCLV in CT cells [28]. In contrast, RVFV piRNAs mapped to both the M and the S segments at similar levels [36]. Interestingly, we saw a clear bias for positive-sense 26–28 nt reads, but a bias for negative-sense 29–30 nt reads that was not observed for either PCLV or RVFV. LACV-derived piRNAs may thus show a size preference depending on whether they are primary or secondary piRNAs. We also saw a substantial peak of positive reads mapping to the LACV S segment at 19 nt, which predominantly represented a single sequence. However, it is unclear why these specific 19 nt sRNA reads are generated in response to LACV infection in Hsu cells. We also observed few LACV-derived siRNA reads that mapped to the LACV M or L segment. It is possible that later timepoints post infection would have resulted in stronger siRNA and/or vpiRNA generation from these segments. It is possibly that increased sRNA targeting of the S segment is related to the overall abundance of the three segments.

We also investigated if PIWI gene silencing affects piRNA or vpiRNA biogenesis. While all PIWI genes seemed to have some impact on piRNA counts, Piwi5 and Piwi6b silencing had the most significant impact on 26–28 nt reads, suggesting that Piwi5 and Piwi6b could be involved in cellular piRNA biogenesis. Piwi5 was previously identified as important for piRNA biogenesis in *Ae. aegypti* [32]. Complete PIWI gene knockout, as well as combinations of knockouts may be required to result in a complete loss of specific piRNA populations. While we observed a proportional increase in 21 nt sRNA reads after silencing of all PIWI genes, we anticipate that this increase is representative of a reduction in overall piRNA abundance, since we normalized all reads of a set length to the total 19–32 nt reads. Thus, a reduction in one sRNA size class can represent as a proportional increase in another sRNA size class. This is a more likely explanation than a specific interaction of all PIWI proteins with the siRNA pathway. However, it was interesting to see that some PIWI genes also decrease the abundance of 22 nt reads, indicative of a reduction in miRNAs. There is no clear link between miRNAs and PIWI genes to our knowledge.

When focusing our analysis on how PIWI gene silencing impacted vpiRNA abundance, we noted that Piwi1, Piwi3, Piwi4, and Piwi6b reduced MERDV-derived positive-sense 26–28 nt reads; however, depending on the gene, silencing did not consistently significantly reduce MERDV-derived reads through 26-28 nt. For negative-sense MERDV-derived sRNA reads, at only Piwi5 silencing significantly reduced MERDV-derived sRNA reads at 29 nt. However, when looking at the summary of all 24–32 nt reads in Figure 7d, it is clear that all PIWI genes are involved in primary or secondary piRNA biogenesis to varying degrees. Piwi5 and Piwi6 have been implicated in the biogenesis of virus-derived piRNAs in *Aedes aegpyti* [32,33]. Piwi6b had the overall biggest impact on both positive-sense and negative-sense 24–32 nt vpiRNA reads, which the impact of Zuc was minimal and restricted to the negative-sense reads. It is possible that Zuc is not involved in piRNA processing in Hsu cells, or that ping-pong-dependent piRNA biogenesis compensates for Zuc-dependent piRNA biogenesis [64]. Silencing PIWI gene expression also significantly increased both positive and negative MERDV-derived siRNA reads, similar to what was seen for total sRNA biogenesis. However, again we attribute this result to the fact that PIWI gene silencing lowered larger sRNA size classes, resulting in a proportional increase in siRNAs.

We focused on Piwi4 silencing during LACV infection to determine whether Piwi4 may impact the generation of LACV-derived sRNAs. There was a significant increase in positive and negative reads at 21 nt when Piwi4 was silenced and, in turn, we observed a small (non-significant) decrease in 26–28 nt positive-sense reads and a more pronounced, significant reduction in 29–30 nt reads derived from the negative-sense strand. Given that there was a larger proportion of positive LACV-derived sRNA reads from 26 to 28 nt and a larger proportion of negative LACV-derived sRNA reads from 29 to 30 nt, this could be why we see Piwi4 silencing affecting these ‘smaller’ positive-sense piRNAs and ‘larger’ negative-sense piRNAs. However, why LACV-derived piRNA biogenesis results in this specific size preference for primary and secondary piRNAs remains unclear. Overall, negative-sense 24–32 nt reads were significantly reduced in Piwi4 silenced cells, indicating that Piwi4 is involved in LACV-derived piRNA biogenesis, but whether this activity is required for its antiviral activity remains unclear, and further experiments using gene knockout, as well as testing for the effect of other PIWI genes on LACV-derived piRNAs are needed. We also suspect some role of Piwi5 during USUV replication that we could not elucidate with only a transient silencing of gene expression and the chosen timepoint post infection. Future studies using gene knockout cells may provide evidence for a role of Piwi5 in controlling USUV replication.

One of the major aims of our study was to try to narrow down why Piwi4 is antiviral. While we saw that Piwi4 silencing impacted vpiRNAs, other PIWI proteins also impacted MERDV vpiRNAs. One potential explanation for Piwi4 antiviral activity was recently suggested to be the indirect effect Piwi4 may have on two abundant piRNAs: tapiR1 and tapiR2 [61,62]. These piRNAs have large impacts on cellular gene regulation in Aag2 cells and it was suggested that their association with Piwi4 could explain indirect antiviral effects of interfering with Piwi4 expression. While we cannot rule this out completely without sequencing PIWI-associated piRNAs or silencing tapiR1 and tapiR2 themselves, we observed significant decreases in both piRNAs after silencing of multiple PIWI genes that have no antiviral effects in Hsu cells. The biggest decrease in tapiR1 and tapiR2 was observed after silencing of Piwi3, which has no antiviral activity in Hsu cells. We thus suggest that the impacts of tapiR1 and tapiR2 alone are unlikely to explain Piwi4 antiviral activity in *Culex* cells.

Finally, we looked at the relationship between vpiRNA biogenesis and the presence of viral DNA forms. Recent research has implicated the presence of vDNA forms as being necessary for maintaining persistent infection in *Drosophila* and mosquitoes [38,65]. In *Aedes* spp., vDNA forms are produced early during virus infection [38,63,66]. Additionally, vDNA forms may serve as a template for virus-derived sRNA biogenesis. Inhibiting vDNA production in Aag2 cells lowered SINV-derived piRNA and siRNA production [37]. We were previously unable to detect any MERDV-derived viral DNA forms in Hsu cells despite abundant vpiRNAs [29]. LACV-derived viral DNA forms have only been identified in *Aedes aegypti* Aag2 cells [63]. We found that LACV viral DNA forms were generated in LACV-infected Hsu cells, but we found that only one short segment of the LACV S segment could be detected as a viral DNA form. While several viral DNA forms spanning the LACV S segment genome were previously detected in LACV-infected Aag2 cells [63], we only identified one viral DNA form in Aag2 cells—the same region that was produced in Hsu cells. The differing findings may be due to different LACV strains or other experimental conditions. Additionally, no PCR product was seen for primer set 6, and a viral DNA form was seen for primer set 7. Primer set 6 uses the same forward primer as primer set 7 and a reverse primer upstream of the reverse primer used in primer set 7. We expected to see a viral DNA form from primer set 6, given that one was observed when using primer set 7. We thought that we did not see a product because viral DNA forms likely come from defective viral genomes [67]. Viral defective genomes have been seen to have a high proportion of mutations or large rearrangements of the genome segments [68], but we sequenced the PCR product generated from infected Hsu DNA with Primer set 7 and it matched our input virus. We hypothesize that the lower binding affinity of primer set 6 (also seen on RNA/cDNA in Figure 10) was responsible for a lack of amplification. This highlights the need to use overlapping primer sets with redundancy when trying to detect DNA forms. We were primarily interested in seeing if viral DNA forms correspond to the detected vpiRNAs. While some LACV-derived vpiRNAs corresponded to the identified vDNA form, many LACV-derived vpiRNAs mapped to regions where no vDNA forms were identified. Our data thus generally support that LACV-derived vpiRNA generation is unlikely to be dependent on the biogenesis of vDNA forms. However, it remains possible that further vDNA forms derived from mutated and/or defective genomes are present in our LACV-infected Hsu cells, and further in-depth sequencing and analysis may be required to identify all present vDNA forms.

In conclusion, we identified that Piwi4 has antiviral activity in *Cx. quinquefasciatus* and *Cx. tarsalis* cells. We detected vpiRNAs derived from LACV for the first time in Hsu cells and found that Piwi4 silencing resulted in a small decrease in those vpiRNAs. However, other PIWI genes also decreased overall piRNAs and MERDV-derived piRNAs in Hsu cells, suggesting a potential overlap and redundancy in PIWI gene function. We found that the antiviral activity of Piwi4 is not likely due to tapiR1 or tapiR2 activity. Finally, we determined that vDNA forms are produced in Hsu cells during LACV infection. Future research should aim to elucidate the process of vpiRNA biogenesis in mosquitoes using gene knockouts, multiple infection timepoints, and a cross-species comparative approach.

## Figures and Tables

**Figure 1 viruses-14-02758-f001:**
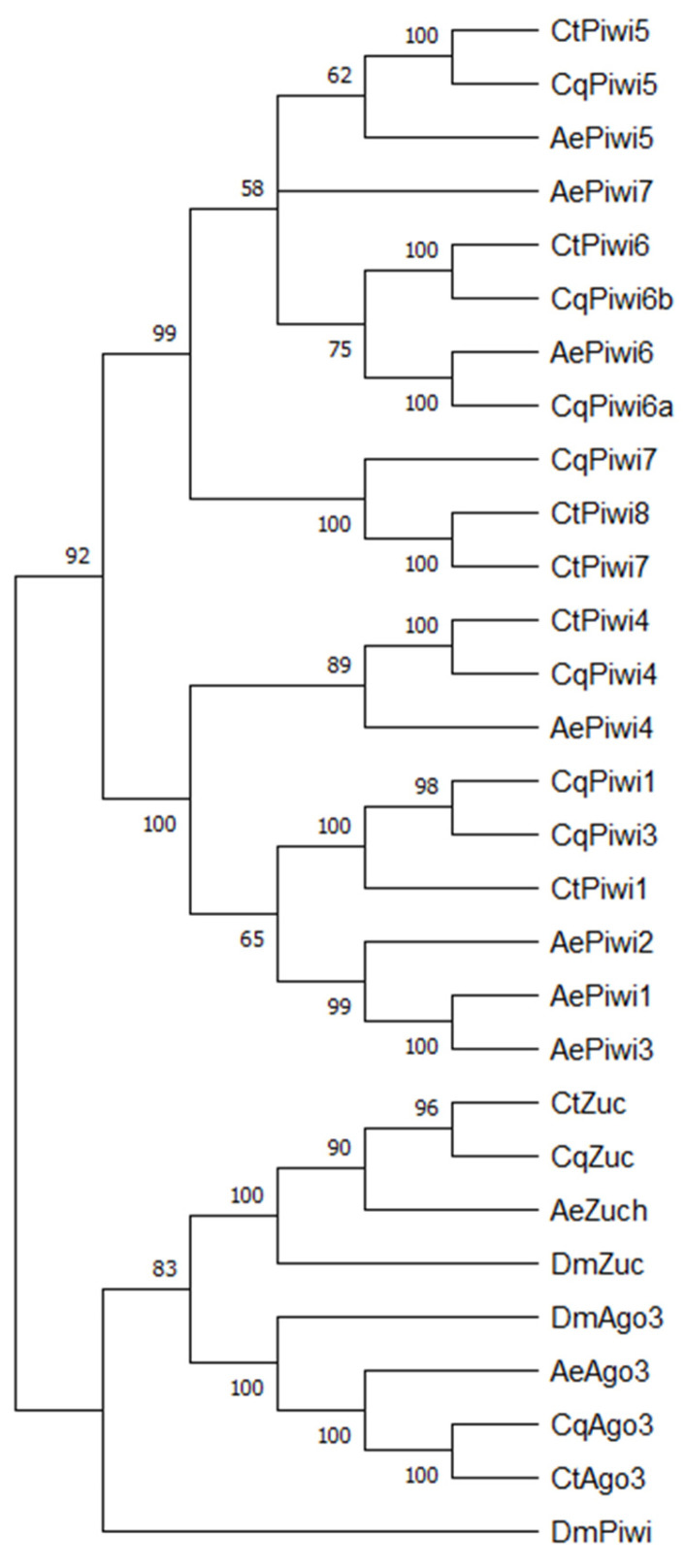
Phylogeny of *Cx. quinquefasciatus*, *Cx. tarsalis*, *Ae. aegypti*, and *D. melanogaster* PIWI family protein sequences. All amino acid sequences were aligned using the MUSCLE algorithm. A maximum likelihood tree was generated using the LG model with 500 bootstrap values. Accession numbers are listed in Appendix A.

**Figure 2 viruses-14-02758-f002:**
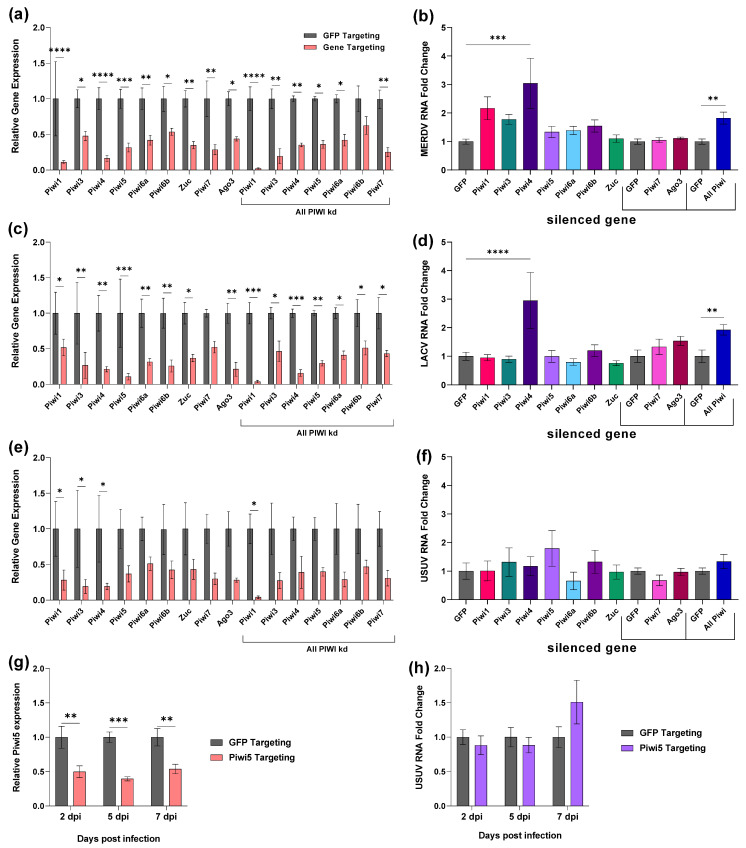
Impact of *Cx. quinquefasciatus* PIWI gene silencing on virus replication. Hsu cells were left uninfected (**a**,**b**) or were infected 2 days post-transfection with LACV MOI 10 (**c**,**d**), USUV MOI 50 (**e**–**h**). Cellular RNA was extracted at 2 dpi (**a**–**f**) or 2, 5, 7 dpi (**g**,**h**). Silencing efficiency of PIWI genes (**a**,**c**,**e**,**g**) and viral RNA levels (**b**,**d**,**f**,**h**) were determined by qRT-qPCR. Bars are the mean +/− SEM of 2 (**g**,**h**) or 3 (**a**–**f**) independent experiments with 3 biological replicates each. Brackets in (**b**,**d**,**f**) indicate that separate experiments were performed for Piwi7 and Ago3, and all PIWI genes, including separate GFP controls. Significant changes in RNA abundance compared to the GFP control are shown as * *p* < 0.05, ** *p* < 0.01, *** *p* < 0.001, **** *p* < 0.0001.

**Figure 3 viruses-14-02758-f003:**
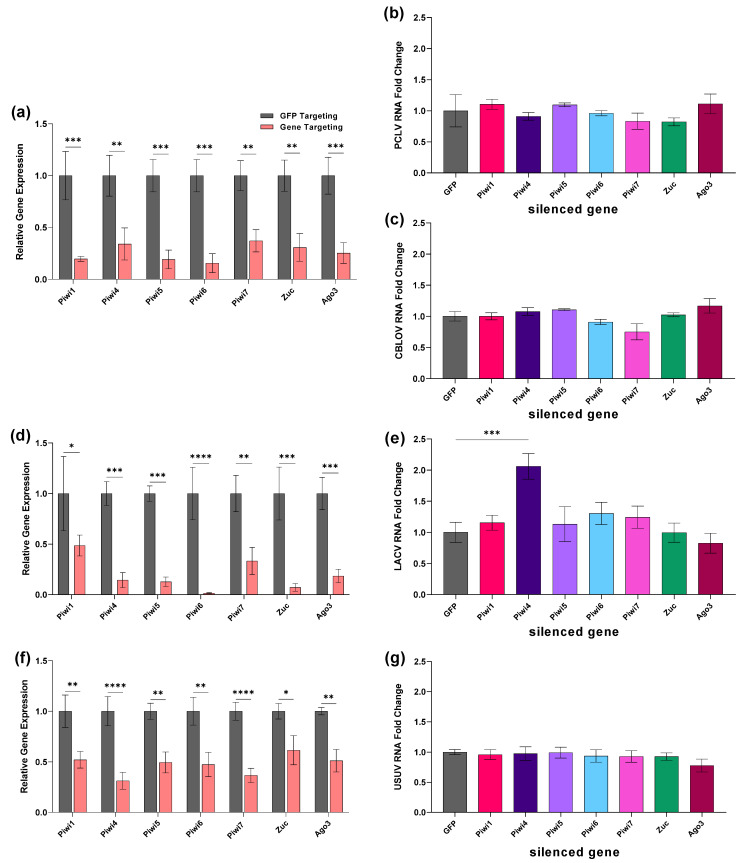
Impact of *Cx. tarsalis piwi* gene silencing on virus replication. CT cells were left uninfected (**a**–**c**) or were infected 2 days post-transfection with either LACV MOI 10 (**d**,**e**), USUV MOI 50 (**f**,**g**). Cellular RNA was extracted at 2 dpi (**a**–**g**). Silencing efficiency of PIWI genes (**a**,**d**,**f**) and viral RNA levels (**b**,**c**,**e**,**g**) were determined by qRT-qPCR. Bars are the mean +/− SEM of 2 independent experiments with 3 biological replicates each. Significant changes in RNA abundance compared to the GFP control are shown as * *p* < 0.05, ** *p* < 0.01, *** *p* < 0.001, **** *p* < 0.0001.

**Figure 4 viruses-14-02758-f004:**
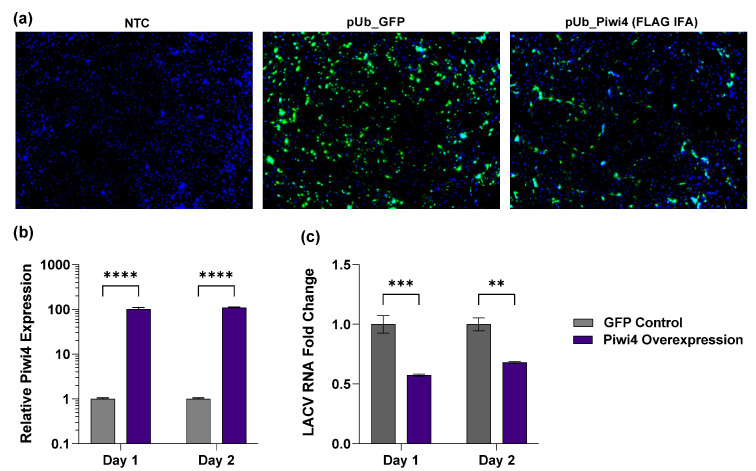
Overexpression of Piwi4 in *Cx. quinquefasciatus* derived cells. (**a**) Immunostaining with anti-FLAG M2 antibodies in Hsu cells transfected with no plasmid (NTC), pUb_GFP (transfection control), or pUb_Piwi4. (**b**) Relative *Cx. quinquefasciatus* Piwi4 gene expression (normalized to the Actin 5c housekeeping gene) was determined in pUb_GFP and pUb_Piwi4-transfected cells. Cells were infected 2 days post-transfection with LACV MOI 10 (**c**). RNA was extracted 1 dpi and 2 dpi. LACV RNA levels were quantified using qRT-PCR and a standard curve normalized to input RNA levels. Significant changes in LACV RNA abundance compared to the GFP control are shown as ** *p* < 0.01, *** *p* < 0.001, **** *p* < 0.0001.

**Figure 5 viruses-14-02758-f005:**
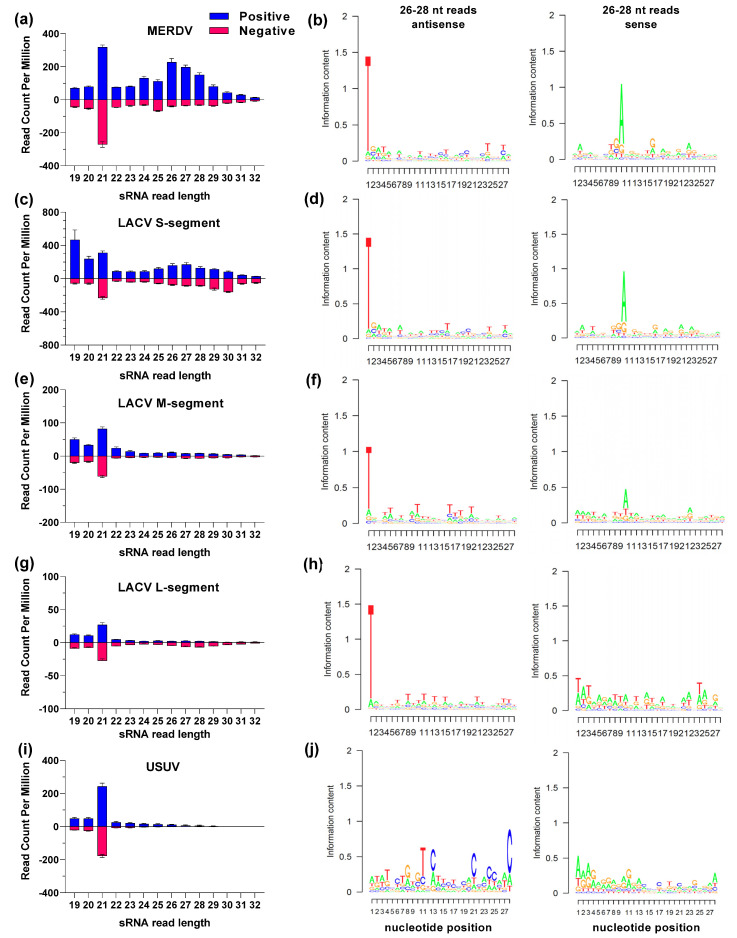
Virus-derived sRNA reads in *Cx. quinquefasciatus* cells. sRNA libraries were sequenced from non-treated Hsu cells (**a**,**b**) or 2 dpi with LACV (**c**–**h**), or USUV (**i**,**j**). 19–32 nt reads were aligned to the respective virus genomes and histograms are shown for MERDV (**a**), LACV S segment (**c**), LACV M segment (**e**), LACV L segment (**g**), and USUV (**i**). The sRNA read counts per million total sRNA reads are shown as the mean of 3 biological replicates. Nucleotide biases of 26–28 nt sRNA reads derived from the antisense (left) and sense (right) strands of MERDV (**b**), LACV S segment (**d**), LACV M segment (**f**), LACV L segment (**h**), and USUV (**j**) are shown.

**Figure 6 viruses-14-02758-f006:**
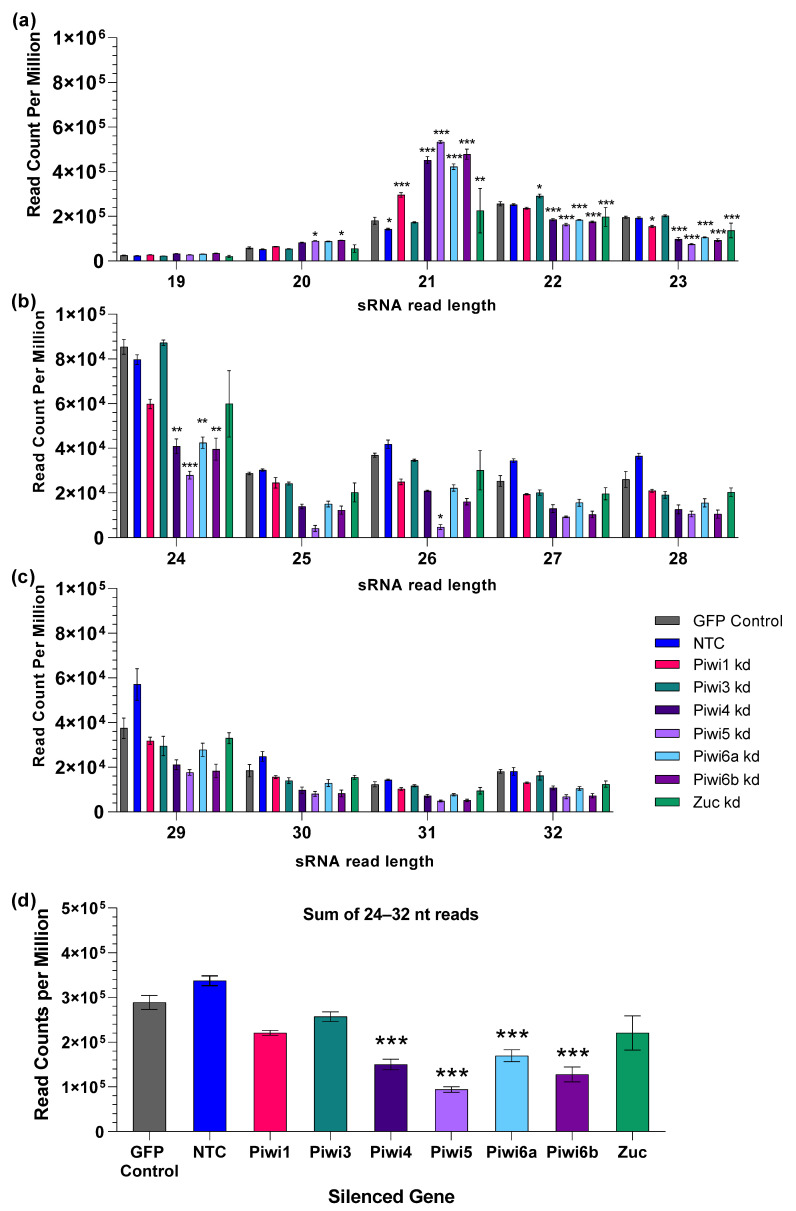
Total small RNAs in *Cx. quinquefasciatus* cells. sRNA libraries were sequenced from non-infected dsRNA-treated cells. All sRNAs (independent of sequence) from 19–32 nt length were analyzed and are plotted separately for better visualization as 19–23 nt (**a**), 24–28 nt (**b**), and 29–32 nt (**c**). Reads were normalized to total sRNA read counts and are shown as reads per million for each size class. A summary of all 24–32 nt read counts, including all potential piRNA size classes is also shown (**d**). GFP dsRNA-treated Hsu cells, and nontreated cells (NTC) were included as negative controls. All sRNA read counts targeting the introduced dsRNA (GFP, Piwi1-6b, Zuc) were removed prior to analysis. Bars represent 3 biological replicates. Significant changes in reads were compared to the GFP control using Two-Way ANOVA (**a**–**c**) or One-Way ANOVA (**d**). Significance is indicated as * *p* < 0.05, ** *p* < 0.01, *** *p* < 0.001.

**Figure 7 viruses-14-02758-f007:**
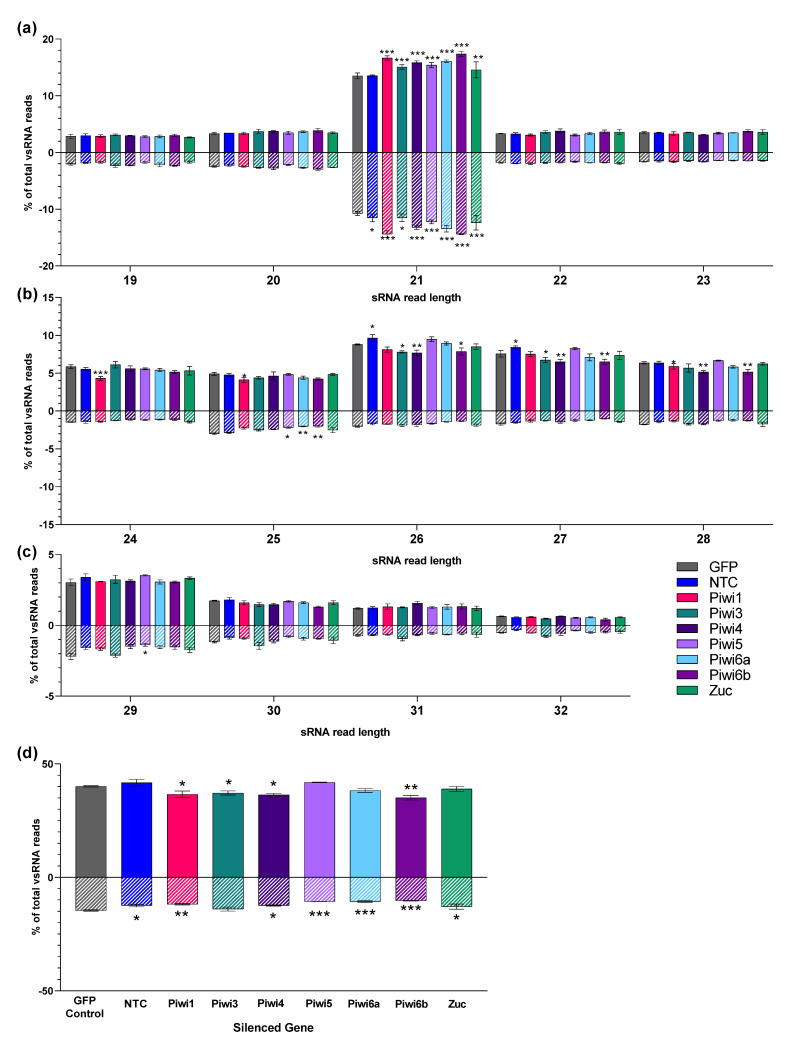
MERDV-derived small RNAs in *Cx. quinquefasciatus* cells. MERDV-derived sRNA reads from 19 to 32 nt length were analyzed and are plotted separately for better visualization as 19–23 nt (**a**), 24–28 nt (**b**), 29–32 nt (**c**), and the sum of all 24–32 nt reads (**d**). Reads from individual size classes were normalized to total MERDV-derived sRNA reads and are shown as % of total MERDV-derived reads for each size class (full bars indicate positive-sense reads, while hatched bars indicate negative-sense reads). GFP dsRNA-treated Hsu cells, and nontreated cells (NTC) were included as negative controls. Bars represent 3 biological replicates. Significant changes in reads were compared to the GFP control using Two-Way ANOVA (**a**–**c**) or One-Way ANOVA (**d**). Significance is indicated as * *p* < 0.05, ** *p* < 0.01, *** *p* < 0.001.

**Figure 8 viruses-14-02758-f008:**
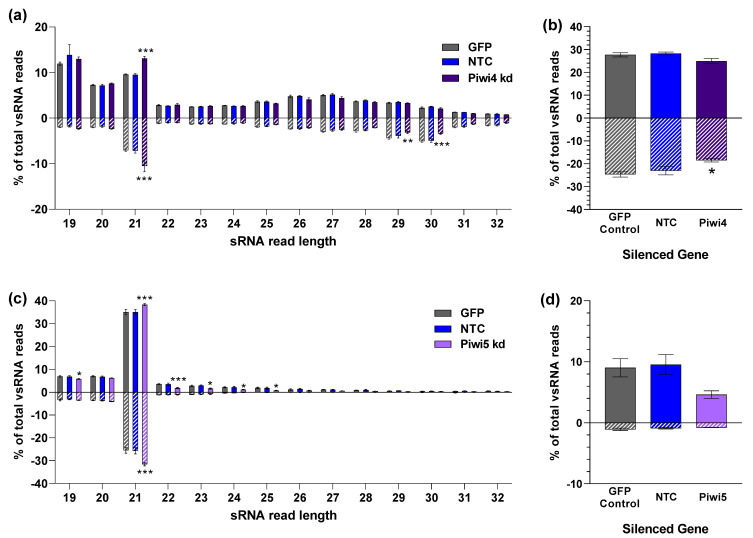
Arbovirus derived small RNAs in *Cx. quinquefasciatus* cells. sRNA libraries were sequenced from Hsu cells 2 dpi with (**a**) LACV MOI 10 or (**b**) USUV MOI 50. 19–32 nt reads were aligned to (**a**) LACV S segment, (**b**) USUV. Cells were treated with Piwi4 dsRNA (**a**), Piwi5 dsRNA (**b**), and GFP dsRNA or no dsRNA (NTC) as negative controls (**a**,**b**). Bars represent 3 biological replicates. Significant changes in reads compared to the GFP control are shown as * *p* < 0.05, ** *p* < 0.01, and *** *p* < 0.001.

**Figure 9 viruses-14-02758-f009:**
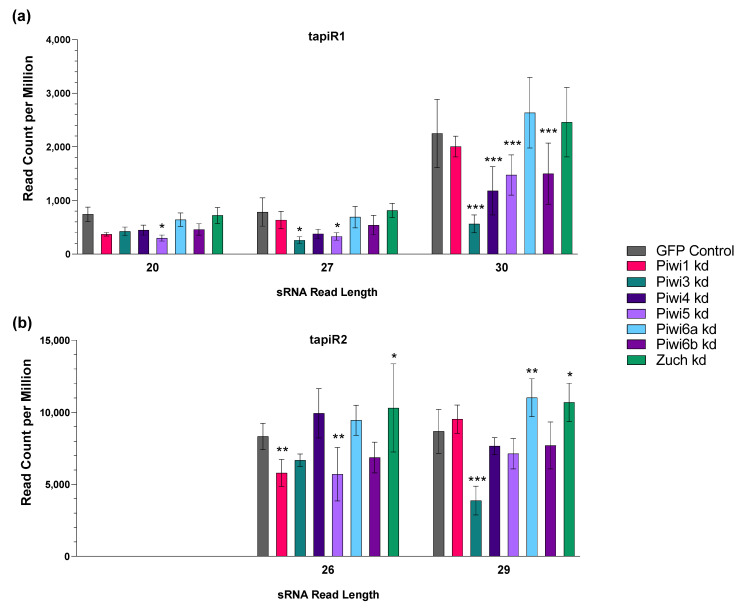
Detection of tapiR1 and tapiR2 in Hsu cells. All sRNA reads from cells treated with PIWI or GFP dsRNA were aligned to tapiR1 (**a**) and tapiR2 (**b**) and the most abundance read lengths were plotted for each piRNA. Significant changes in reads compared to the GFP control are shown as* *p* < 0.05, ** *p* < 0.01, and *** *p* < 0.001.

**Figure 10 viruses-14-02758-f010:**
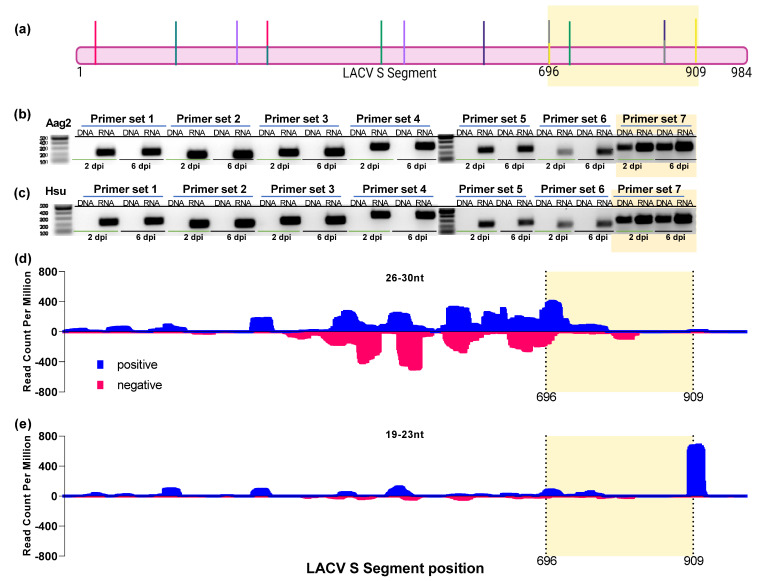
Generation of viral DNA forms in mosquito cells following LACV infection. PCR was performed using genomic DNA, and cDNA from LACV-infected Aag2 and Hsu cells. (**a**) Overlapping primer sets covering the LACV S segment were designed. Colors indicate primer pairs used to amplify specific regions. (**b**) PCR products from Aag2 LACV-infected cells at 2 dpi or 6 dpi. (**c**) PCR produced from Hsu LACV-infected cells at 2 dpi or 6 dpi. (**d**) Positional mapping of LACV-derived piRNAs along the LACV S segment. (**e**) Positional mapping of LACV-derived 26–30 nt (**d**) and 19–23 nt (**e**) sRNAs along the LACV S segment. Yellow highlighted regions (**a**–**e**) indicate where LACV vDNA was produced.

**Table 1 viruses-14-02758-t001:** *Cx. quinquefasciatus* (Cq) and *Cx. tarsalis* (Ct) primers used to generate dsRNA.

Gene	Accession Numbers	dsRNA Primers
CqPiwi1	XM_001844015.2	F-taatacgactcactatagggCCACGATCGCAACTACATGG
		R-taatacgactcactatagggTTCCGGAAGTGAATCGACCA
CqPiwi3	XM_038255983.1	F-taatacgactcactatagggATCGGGGACACTCTTCGAAC
		R-taatacgactcactatagggCACACGAGAATGTCCTGCTC
CqPiwi4	XM_038251629.1	F-taatacgactcactatagggATAGCAGTGAGGGTCGTGAC
		R-taatacgactcactatagggCTCAGCTGGTAAACATCGCC
CqPiwi5	XM_038251859.1	F-taatacgactcactatagggCACTACCAAGCTGAGCATGC
		R-taatacgactcactatagggGTGCCAACCTTACGCAACTT
CqPiwi6a	XM_038251857.1	F-taatacgactcactatagggCCGACGCAGGTAATCAAGTG
		R-taatacgactcactatagggCAATCTTGTCCCTGATGGCG
CqPiwi6b	XM_038258148.1	F-taatacgactcactatagggCGGAGGTTATCAACATGGCG
		R-taatacgactcactatagggTCGCACAGCTTGTTCCTAGA
CqPiwi7	XM_038266499.1	F-taatacgactcactatagggTGCAGAGCCAGCAGGATTAC
		R-taatacgactcactatagggCTCGGATCCCGAATGACGAT
CqZuc	XM_001870711.2	F-taatacgactcactatagggGTTTCGTGCTGTTTTCCGA
		R-taatacgactcactatagggTTGGCCCGTATCAACGCGTC
CqAgo3	XM_038254040.1	F-taatacgactcactatagggAATCTGGACGTTTCGCATCG
		R-taatacgactcactatagggCGCCTCCTTGTTTTGGTTGA
CtPiwi1	mRNA6102 *	F-taatacgactcactatagggCGAATTGATGGCCCTGGTTC
		R-taatacgactcactatagggCCATGACCTGCGTTGGAATC
CtPiwi4	mRNA2582 *	F-taatacgactcactatagggCCATCCGCGAGTACCAGATT
		R-taatacgactcactatagggCGGCGATTGAATGTCTCCAA
CtPiwi5	mRNA13996 *	F-taatacgactcactatagggCTTGTCCGTCCCGTTGAAGA
		R-taatacgactcactatagggCCTACTCCATCCCGGTAGA
CtPiwi6	mRNA4897 *	F-taatacgactcactatagggACGAGGAATGGCCCGATAAC
		R-taatacgactcactatagggCAAACAGCCCAAGAACTGGC
CtPiwi7	mRNA5433 *	F-taatacgactcactatagggTGCCGGAGATTGAGAGTGT
		R-taatacgactcactatagggATTCCGTAGCAGGTGTCC
CtZuc	mRNA10624 *	F-taatacgactcactatagggAGGGCAGTGCCTACTTGATG
		R-taatacgactcactatagggCGTCACGATGATGTTGTCCC
CtAgo3	mRNA4166 *	F-taatacgactcactatagggCTGAGCCAGTCCAGCTCATC
		R-taatacgactcactatagggCCGCTTGACCGGACTCTTTT

* Internal identifier—sequences included in Appendix A.

**Table 2 viruses-14-02758-t002:** *Cx. quinquefasciatus* (Cq), *Cx. tarsalis* (Ct), and virus primers for RT-qPCR.

Gene	Accession Numbers	qPCR Primers	Reference
CqPiwi1	XM_001844015.2	F-GCAGCTGACCAGCAACTATTTC	[29]
		R-CCCAAACGTCTTCTTGTGTTCC	
CqPiwi3	XM_038255983.1	F-CTGGTCGGACGTAATCTGTTTGA	[29]
		R-CATCGTCTTGTGCGCGATTTC	
CqPiwi4	XM_038251629.1	F-TTTCCAACTACCTCCCGATCAAC	[29]
		R-CGCCATCACGGTAGAAGATGATAC	
CqPiwi5	XM_038251859.1	F-TGAAGTTGACGCTGATTGGG	[29]
		R-ACGATGGGTAAGTTCTGCAC	
CqPiwi6a	XM_038251857.1	F-CTACATTACCAG-CATCCGACAG	[29]
		R-TGCACTTCTCAAACAGGTCG	
CqPiwi6b	XM_038258148.1	F-TCAAGGTGCTCATGGAATCG	[29]
		R-GACCGTTGAGTAGAATTCCGAG	
CqPiwi7	XM_038266499.1	F-CGGAAACTGGCGTAATGGTA	-
		R-TTCGTTCAACTGCGGACTAC	
CqZuc	XM_001870711.2	F-TACATCGTGACGGTG-GACAAG	[29]
		R-GAACTGTACGCCATCGAGGAA	
CqAgo3	XM_038254040.1	F-AGTACATCAACCAGCATCGAG	[29]
		R-TGCAGAATTGTTTCCACGTTG	
CtPiwi1	mRNA6102 *	F-ATGGTAAACAA-GCTCCGTAGTG	-
		R-GTTCCGGTGTGGACAATCTT	
CtPiwi4	mRNA2582 *	F-ACGGCAAACGGAGTACAA	-
		R-GATCGGTTGGGTACGATGAA	
CtPiwi5	mRNA13996 *	F-GACGCTGATTGGGAGAAACTA	-
		R-GTCACATAGCCCGGGTATAAAT	
CtPiwi6	mRNA4897 *	F-GCCATCAGGGACAAGATTGA	-
		R-ATACGCTTGCTCACGACTATG	
CtPiwi7	mRNA5433 *	F-CCATCCTAGCGAA-GCTCAAA	-
		R-GGAAGAG-TCGGGTGTTGATG	
CtZuc	mRNA10624 *	F-TCGGCATGTACATCGTGAC	-
		R-TCTTCCTCAGCCAACTTTACC	
CtAgo3	mRNA4166 *	F-CACTCGAATGTTCCG-GATTGA	-
		R-CAC-TCGAATGTTCCGGATTGA	
CqActin 5c	XM_038249510.1	F-CAACTGCCCAAATCGAATGAC	-
		R-CGACGCACTCTCGGAATAAA	
CtActin	GU390398	F-GACTACCTGATGAAGATCCTGAC	[55]
		R-GCACAGCTTTTCCTTGATGTCGC	
MERDV	MH310083.1	F-CCTCCTCCCTCCGCTCTAGTT	-
		R-CGGCTTACAACTTGGCTCTC	
LACV (L-segment)	OP962744	F-CAGCCCAGACAGCCATAAA	-
		R-CCCTGGTAGCATGTTGTATGT	
USUV	MT188658.1	F-CATCAAGGTTCTCTGCCCATAC	-
		R-GAAAGAGGGACTCGAACCAATC	
CLBOV	KX669689.1	F-TGGACGTGGCTTGTTTTATCGC	-
		R-GCGCCAGAGCATAGCAATGTAG	
PCLV (S-segment)	KU936055.1	F-AGGACTTGATGTTCTCGGTATT	-
		R-GATCATAGTGCTCACGTCATTCT	

* Internal identifier—sequences included in Appendix A.

**Table 3 viruses-14-02758-t003:** PCR Primers for LACV S segment (OP962746) vDNA Detection.

Set	Sequence
Set 1	F-TTTTTTACCTAAGGGGAAAT
	R-GCCTTCCTCTCTGGCTTACG
Set 2	F-TGATGTCGCATCAACAGGTG
	R-GCCTTCCTCTCTGGCTTACG
Set 3	F-AGCCAGAGAGGAAGGCTAACC
	R-AGTTGTCCTGATCAACTCG
Set 4	F-CAGGACAACTATTATCAACC
	R-AGCTGCTCTACATCCTTCAGG
Set 5	F-ACGCTATGGCACTCTCACAG
	R-TTGACATATATAAATTTAGAAT
Set 6	F-CCTGAAGGATGTAGAGCAGCTT
	R-TTGACATATATAAATTTAGAAT
Set 7	F-CCTGAAGGATGTAGAGCAGCTT
	R-ACCCATTTAGCTGCTATTT
CqActin	F-CAACTGCCCAAATCGAATGAC
	R-CGACGCACTCTCGGAATAAA
AeActin	F-GAATGTGCAAGGCCGGATTC
	R-GCTCGATCGGGTACTTCAGG

**Table 4 viruses-14-02758-t004:** Average sequencing read counts of all 19–32 nt reads in *Cx. quinquefasciatus* cells.

Sample	Total Reads	MERDV Reads	MERDV Read %	LACV Reads	LACV Read %	USUV Reads	USUV Reads %
MERDV no dsRNA	6.51 M	15,647	0.23%	-	-	-	-
MERDV GFP	12.1 M	18,341	0.15%	-	-	-	-
MERDV Piwi1	12.4 M	14,661	0.11%	-	-	-	-
MERDV Piwi3	29.2 M	22,759	0.08%	-	-	-	-
MERDV Piwi4	20.1 M	17,886	0.09%	-	-	-	-
MERDV Piwi5	28.4 M	20,582	0.07%	-	-	-	-
MERDV Piwi6a	37.3 M	34,895	0.09%	-	-	-	-
MERDV Piwi6b	21.7 M	15,501	0.07%	-	-	-	-
MERDV Zuc	13.3 M	14,187	0.10%	-	-	-	-
LACV no dsRNA	28.5 M	104,271	0.38%	90,795	0.39%	-	-
LACV GFP	11.5 M	43,156	0.37%	44,032	0.37%	-	-
LACV Piwi4	8.05 M	42,571	0.52%	39,469	0.49%	-	-
USUV no dsRNA	13.9 M	55,426	0.39%	-	-	10,604	0.07%
USUV GFP	7.07 M	28,267	0.40%	-	-	4976	0.07%
USUV Piwi5	8.10 M	35,825	0.45%	-	-	9393	0.11%

## Data Availability

All small RNA sequencing data presented in this study are openly available in the NCBI SRA database under BioProject PRJNA898416.

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
