# Peer review of "Culex Mosquito Piwi4 Is Antiviral against Two Negative-Sense RNA Viruses"

_viruses, 2022, doi:10.3390/v14122758_

Round 1
Reviewer 1 Report
This is a very well-written and presented paper. The discussion is well-rounded, the methods are described in good detail (I particularly appreciate that the history of each virus stock was included) and the results have been written in a very transparent manner.
I mostly have a few suggestions which may assist readers who are not familiar with the field and overall improve interpretation of the results.
Introduction:
1. The background information provided in the introduction is excellent but due to the complex topic it is a bit difficult to follow. It would be great to have a summary table or figure to help the reader understand the current literature on piRNA pathways in mosquitoes.
2. Line 130 – this line needs to be reworded
3. Line 132 - It would be good to outline in the introduction what family each of the viruses studied belongs to, for those who are not familiar with these particular viruses, and to highlight the breadth of the study.
Results:
1. Line 363 – Although the authors hypothesise that CtPiwi8 is likely a sequencing artifact, for accessibility and since the sequence is included in the phylogenetic tree it would be best to also include this sequence in supplementary table 1, include an alignment with ctPiwi7 or just include the sequence in the supplementary files
2. Figure 2b – it would be good to include x-axis label to indicate that genes on the x-axis are knocked down or label "gene kd" as for figure 6
3. Line 441 – 443 : Authors state a high proportion of cells expressed the Piwi4 plasmid but I don’t think the immunostaining in figure 4a particularly supports this statement, especially compared to the GFP plasmid.
4. Figure 5 - I would suggest adding the virus (and genome segment) name to the figures to help with interpretation. Similarly it would be helpful to indicate on the figure that the graphs of nucleotide biases are for 26-28 nt sRNA reads.
5. Figure 5 - in the part of the legend referring to the histograms, I think the figures for LACV S and M segment should read be (c) and (e) rather than (b) and (d) (see text pasted below):
“19-32nt reads were aligned 508 to the respective virus genomes and histograms are shown for MERDV (a), LACV S segment (b), 509 LACV M segment (d), LACV L segment (g), and USUV (i).”
6. Line 480 – use “were” instead of “was”
7. Statements at lines 573 and 581 are in contrast with each other.
8. Figure 7 – indicate in figure legend that hatched bars represent negative-sense reads
9. Line 609 –would be good to include the data in the supplementary figure to support the statement on positional mapping of MERDV siRNA and piRNA reads in this study
10. Figure 7 is very convincing and the discussion on why Primer 7 amplified product while primer set 6 did not is good. It would be interesting to know if the authors sequenced this PCR product and if the sequence confirms their theory about mutations in the region where the reverse primer of set 6 binds.
11. Line 804 – typographical error in the word “Impact”
Author Response
General:
In addition to the below reviewer-specific edits, we have added the SRA database BioProject number for our uploaded sequencing raw data (PRJNA898416) and an additional supplemental file (Supplemental File 2) with all Culex tarsalis PIWI coding sequences.
The accession numbers for LACV are in progress (there was a technical hiccup during the first BankIT submission, so we submitted them again and are just waiting for NCBI to process the resubmitted sequences and assign an accession number, which will then be placed in the relevant tables).
Going through our data again we noticed a few new observations that have been added/adjusted:
For the analysis in Figure 6, we filtered out all reads mapping to the respective dsRNA, as brought up by reviewer 3. As we did this, we decided to also add another panel to Figure 6-8 that would show the summarized 24-32nt read counts following PIWI knockdown. It better captures the overall impact on the sum of piRNAs. We’ve appropriately adjusted the text of these sections to include this information and figure references.
After a deep dive back into our data, we now noticed that the peak in 19nt and 20nt reads previously shown for Piwi1 silencing in Figure 7a was due to two specific reads in these samples (TACATGGACTCGTTCAAGC and TACATGGACTCGTTCAAGCA). We realized that these reads are actually derived from our Piwi1 dsRNA. These reads mapped to Merida virus with our allowed 2 mismatches. No significant number of 21nt or larger sizes mapped to the Piwi1 dsRNA sequence. These are obviously very small odds to have overlap between our viral genome and the introduced dsRNA. We always check for host-off targets but not necessarily for the virus.
In the revised version, we removed all reads mapping to Piwi1 dsRNA from the MERDV-mapped reads. We checked the other PIWI knockdown samples and these did not contain these overlapping dsRNA/virus reads. It has taught us a valuable lesson of filtering out all dsRNA reads when analyzing similar samples in the future and to check our dsRNA also against the viruses we use.
Due to the changes in the figures, some of the writing needed to be adjusted (mostly sections relating to figures 6-8). The overall conclusions remain the same/similar, but minor changes in significance needed to be adjusted and the newly added panels needed to be described. We made a few additional minor editorial changes (all changes tracked), simply to improve clarity or writing.
Reviewer 1:
This is a very well-written and presented paper. The discussion is well-rounded, the methods are described in good detail (I particularly appreciate that the history of each virus stock was included) and the results have been written in a very transparent manner.
We thank the reviewer for the kind words, their time to review our manuscript, and providing the below feedback to improve our paper.
Line numbers provided with the responses refer to the marked up (track changes) revised manuscript.
I mostly have a few suggestions which may assist readers who are not familiar with the field and overall improve interpretation of the results.
Introduction:
- The background information provided in the introduction is excellent but due to the complex topic it is a bit difficult to follow. It would be great to have a summary table or figure to help the reader understand the current literature on piRNA pathways in mosquitoes.
We appreciate the complexity of the topic. We opted to add a graphical abstract instead of a background figure – we hope this works for the reviewer. However, it was hard to make a ‘perfect’ summary due to the nature of having so many open questions and contradictory publications in the literature. We tried to capture the important aspects in this Graphical abstract.
- Line 130 – this line needs to be reworded
We have reworded the statement for clarity in new line 135-139.
- Line 132 - It would be good to outline in the introduction what family each of the viruses studied belongs to, for those who are not familiar with these particular viruses, and to highlight the breadth of the study.
We have added this information in lines 130-135 of the revised manuscript.
Results:
- Line 363 – Although the authors hypothesise that CtPiwi8 is likely a sequencing artifact, for accessibility and since the sequence is included in the phylogenetic tree it would be best to also include this sequence in supplementary table 1, include an alignment with ctPiwi7 or just include the sequence in the supplementary files
We have added a supplementary file with all Culex tarsalis PIWI sequences, including CtPiwi8. We also added an alignment of CtPiwi7 and CtPiwi8 at the end of this supplemental file, showing 97.3 % conserved nucleotide sequence identity between the two. We are not sure if the existence of the two sequences is an artifact, it could be real, but it makes distinction between the two PIWI genes difficult (and as we said, we used primers binding in the regions that are distinct for Piwi8 and were unable to detect it in our CT cells). Any generated dsRNA would likely also target both genes.
- Figure 2b – it would be good to include x-axis label to indicate that genes on the x-axis are knocked down or label "gene kd" as for figure 6
Thank you for the comment. Since it is hard to fit the ‘kd’ behind every gene (as the figures are so busy already), we decided to add the x-axis label ‘Silenced gene’ to figures 2b,d,f, as well as Figure 3b,c,e,g and some of the other figures. We hope this is ok with the reviewer.
- Line 441 – 443: Authors state a high proportion of cells expressed the Piwi4 plasmid but I don’t think the immunostaining in figure 4a particularly supports this statement, especially compared to the GFP plasmid.
We agree with the reviewer that the immunostaining in Figure 4c shows less signal than the GFP expressing transfection control. The IFA for Piwi4 results in a weaker signal in some cells and a brighter signal in others, making it harder to visualize within the MDPI formatting limitations (in text Figure). Also, the use of ‘high proportion’ is of course a subjective term. With mosquito cells, we are often happy to get 30-40% transfection efficiency. We adjusted the wording of this sentence to simply state that we were able to detect FLAG-tagged Piwi4 (Lines 452-453). Any quantitative measure is provided by the qRT-PCR results in Figure 4c.
- Figure 5 - I would suggest adding the virus (and genome segment) name to the figures to help with interpretation. Similarly it would be helpful to indicate on the figure that the graphs of nucleotide biases are for 26-28 nt sRNA reads.
We added the suggested labels for clarity.
- Figure 5 - in the part of the legend referring to the histograms, I think the figures for LACV S and M segment should read be (c) and (e) rather than (b) and (d) (see text pasted below):
“19-32nt reads were aligned 508 to the respective virus genomes and histograms are shown for MERDV (a), LACV S segment (b), 509 LACV M segment (d), LACV L segment (g), and USUV (i).”
Thank you for catching this mistake. We have corrected the figure legend accordingly.
- Line 480 – use “were” instead of “was”
Thank you, we corrected this grammar error.
- Statements at lines 573 and 581 are in contrast with each other.
Thank you for catching this. We removed ‘but not Zuc’ (new line 595) from the first sentence, since this is an incorrect statement.
- Figure 7 – indicate in figure legend that hatched bars represent negative-sense reads
We have added this statement into the legend (new line 632-633).
- Line 609 –would be good to include the data in the supplementary figure to support the statement on positional mapping of MERDV siRNA and piRNA reads in this study
Thank you for this comment. We had these included in one of our drafts and then felt it was redundant to our previous publication (since none of the conditions showed anything different to what we had shown before). However, we can appreciate that it is better to see the data to confirm that there is no difference between the silencing conditions. We have added this back in as Supplemental Figures 2 (19-23nt) and 3 (26-30 nt). We also adjusted the text to place this statement with the corresponding section on MERDV (instead of having it with the LACV data), see lines 620-625.
- Figure 7 is very convincing and the discussion on why Primer 7 amplified product while primer set 6 did not is good. It would be interesting to know if the authors sequenced this PCR product and if the sequence confirms their theory about mutations in the region where the reverse primer of set 6 binds.
We had not previously sequenced the fragment, but we did now. It does actually contain the correct region for the reverse primer 6 without mutation and overall matches our input virus. Primer set 6, although designed as all the other primers, may simply have been a less sensitive primer pair. It showed weaker bands from the cDNA in Figure 10 and may have less binding affinity than primer set 7. While this represents a technical issue more than biological, we left it in, as is, to highlight the importance of multiple/overlapping primer sets to avoid missing DNA forms. We have also adjusted our discussion appropriately to include this information (lines 895-899).
- Line 804 – typographical error in the word “Impact”
Thank you, we corrected this typo (and also made it past tense, which was more appropriate here).
Reviewer 2 Report
Remarks to the Author:
Mosquitoes transmit many viruses inducing the widespread spread of relevant virus in the population. So, it is important for understand how virus can infect mosquitoes and why no pathology in mosquitoes. The authors found Piwi4 is an antiviral gene in Cx. Quinquefasciatus and Cx. Tarsalis cells. Piwi4 inhibited LACV in Hsu and CT cells by producing piRNAs but not USUV infection in Hsu cells. However, why piRNA induced by LACV infection through piwi4 had a role on virus infection, in another word, which piRNA was essential for suppressing virus infection is unknown. Therefore, the antiviral mechanism of piwi4 is unclear.
Major points:
Are those shRNA Hsu cells stable or only transient transfection? Cause those cells need to be infected by virus for several days?
Only RNA fold changes to determine virus level is not enough.
Can you check the target of these virus-derived sRNA sequences?
Overexpression of several of virus-derived sRNA effects viral infection or not?
Author Response
General:
In addition to the below reviewer-specific edits, we have added the SRA database BioProject number for our uploaded sequencing raw data (PRJNA898416) and an additional supplemental file (Supplemental File 2) with all Culex tarsalis PIWI coding sequences.
The accession numbers for LACV are in progress (there was a technical hiccup during the first BankIT submission, so we submitted them again and are just waiting for NCBI to process the resubmitted sequences and assign an accession number, which will then be placed in the relevant tables).
Going through our data again we noticed a few new observations that have been added/adjusted:
For the analysis in Figure 6, we filtered out all reads mapping to the respective dsRNA, as brought up by reviewer 3. As we did this, we decided to also add another panel to Figure 6-8 that would show the summarized 24-32nt read counts following PIWI knockdown. It better captures the overall impact on the sum of piRNAs. We’ve appropriately adjusted the text of these sections to include this information and figure references.
After a deep dive back into our data, we now noticed that the peak in 19nt and 20nt reads previously shown for Piwi1 silencing in Figure 7a was due to two specific reads in these samples (TACATGGACTCGTTCAAGC and TACATGGACTCGTTCAAGCA). We realized that these reads are actually derived from our Piwi1 dsRNA. These reads mapped to Merida virus with our allowed 2 mismatches. No significant number of 21nt or larger sizes mapped to the Piwi1 dsRNA sequence. These are obviously very small odds to have overlap between our viral genome and the introduced dsRNA. We always check for host-off targets but not necessarily for the virus.
In the revised version, we removed all reads mapping to Piwi1 dsRNA from the MERDV-mapped reads. We checked the other PIWI knockdown samples and these did not contain these overlapping dsRNA/virus reads. It has taught us a valuable lesson of filtering out all dsRNA reads when analyzing similar samples in the future and to check our dsRNA also against the viruses we use.
Due to the changes in the figures, some of the writing needed to be adjusted (mostly sections relating to figures 6-8). The overall conclusions remain the same/similar, but minor changes in significance needed to be adjusted and the newly added panels needed to be described. We made a few additional minor editorial changes (all changes tracked), simply to improve clarity or writing.
Reviewer 2:
Mosquitoes transmit many viruses inducing the widespread spread of relevant virus in the population. So, it is important for understand how virus can infect mosquitoes and why no pathology in mosquitoes. The authors found Piwi4 is an antiviral gene in Cx. Quinquefasciatus and Cx. Tarsalis cells. Piwi4 inhibited LACV in Hsu and CT cells by producing piRNAs but not USUV infection in Hsu cells. However, why piRNA induced by LACV infection through piwi4 had a role on virus infection, in another word, which piRNA was essential for suppressing virus infection is unknown. Therefore, the antiviral mechanism of piwi4 is unclear.
Major points:
Are those shRNA Hsu cells stable or only transient transfection? Cause those cells need to be infected by virus for several days?
This is a transient transfection with dsRNA. However, our knockdown validations shown in Figures 2 and 3 (as well as validations we did prior to sRNA sequencing) were all done at the time of sampling, suggesting that the knockdown was stable enough over the duration of virus infection. Knockdown efficiency may have been even greater at the time of infection, but it was still significant at the time of sampling as shown in the respective figures (2 and 3).
Only RNA fold changes to determine virus level is not enough.
We appreciate this comment, and we are aware that many researchers would choose to show plaque assay data to test for infectious virus in the supernatant. However, it is also common in mosquito cells to use intracellular RNA or luciferase assay-based reporter systems to measure virus replication without measuring released infectious virus.
(See for example: Scherer et al. 2021 – doi:10.3390/v13061066; Fragkoudis et al. 2008 – doi:10.1111/j.1365-2583.2008.00834.x; Machado et al. 2022 uses only RNA/luciferase for SINV – doi:10.1371/journal.ppat.1010694;).
In addition, Hsu cells do not produce high levels of virus in the supernatant, which is why we opted for the more sensitive method of detecting intracellular viral RNA. In future studies, where we will further investigate Piwi4 in detail and in vivo, we will use plaque assay data as well as RNA to further characterize where/how this antiviral protein impacts virus replication and production. However, due to the consistency in our data and with what is known for Aedes aegypti cells, we believe that we can conclude that Piwi4 is antiviral in our data-set from this RNA-based data.
Can you check the target of these virus-derived sRNA sequences?
The reviewer did not reference which virus-derived sRNA sequences they are referring to, so it is hard to sufficiently respond. In Supplemental Figure 2, we show where the sRNAs targeting the LACV S genome and based on a comment by reviewer 1, we have added another supplemental figure with the MERDV targeting data (new Supplemental Figures 2 and 3).
Overexpression of several of virus-derived sRNA effects viral infection or not?
This is a very complicated experiment – we have discussed this with collaborators, and it seems very hard to replicate what happens during infection with externally introduced sRNAs. These experiments are rarely done and rarely conclusive (they can be, but not easily). I am also not sure how it would help us here in the context of our research question. Possibly, testing vpiRNA transfection into Hsu cells where Piwi4 is silenced (and in a control) to see if vpiRNAs can reduce virus replication and if this is lost after Piwi4 silencing. However, we think this is outside the scope of this already extensive study.
Reviewer 3 Report
Very well-designed study. Wealth of useful data. I recommend publishing this manuscript after one minor concern is addressed.
Authors mention that "However, when focusing on 21 nt reads (Figure 6a), we noted that Hsu cells not treated with any dsRNA had the least amount of 21 nt siRNAs. We anticipate that this is due to the high abundance of 21 nt siRNAs derived from the dsRNA treatment." Did the authors filter out small RNAs mapping on to the dsRNA that they transfected? Not doing so may change the cpm values, decreasing them because some of the sequencing space is taken up by the dsRNA-derived reads.
Author Response
General:
In addition to the below reviewer-specific edits, we have added the SRA database BioProject number for our uploaded sequencing raw data (PRJNA898416) and an additional supplemental file (Supplemental File 2) with all Culex tarsalis PIWI coding sequences.
The accession numbers for LACV are in progress (there was a technical hiccup during the first BankIT submission, so we submitted them again and are just waiting for NCBI to process the resubmitted sequences and assign an accession number, which will then be placed in the relevant tables).
Going through our data again we noticed a few new observations that have been added/adjusted:
For the analysis in Figure 6, we filtered out all reads mapping to the respective dsRNA, as brought up by reviewer 3. As we did this, we decided to also add another panel to Figure 6-8 that would show the summarized 24-32nt read counts following PIWI knockdown. It better captures the overall impact on the sum of piRNAs. We’ve appropriately adjusted the text of these sections to include this information and figure references.
After a deep dive back into our data, we now noticed that the peak in 19nt and 20nt reads previously shown for Piwi1 silencing in Figure 7a was due to two specific reads in these samples (TACATGGACTCGTTCAAGC and TACATGGACTCGTTCAAGCA). We realized that these reads are actually derived from our Piwi1 dsRNA. These reads mapped to Merida virus with our allowed 2 mismatches. No significant number of 21nt or larger sizes mapped to the Piwi1 dsRNA sequence. These are obviously very small odds to have overlap between our viral genome and the introduced dsRNA. We always check for host-off targets but not necessarily for the virus.
In the revised version, we removed all reads mapping to Piwi1 dsRNA from the MERDV-mapped reads. We checked the other PIWI knockdown samples and these did not contain these overlapping dsRNA/virus reads. It has taught us a valuable lesson of filtering out all dsRNA reads when analyzing similar samples in the future and to check our dsRNA also against the viruses we use.
Due to the changes in the figures, some of the writing needed to be adjusted (mostly sections relating to figures 6-8). The overall conclusions remain the same/similar, but minor changes in significance needed to be adjusted and the newly added panels needed to be described. We made a few additional minor editorial changes (all changes tracked), simply to improve clarity or writing.
Reviewer 3:
Very well-designed study. Wealth of useful data. I recommend publishing this manuscript after one minor concern is addressed.
Authors mention that "However, when focusing on 21 nt reads (Figure 6a), we noted that Hsu cells not treated with any dsRNA had the least amount of 21 nt siRNAs. We anticipate that this is due to the high abundance of 21 nt siRNAs derived from the dsRNA treatment." Did the authors filter out small RNAs mapping on to the dsRNA that they transfected? Not doing so may change the cpm values, decreasing them because some of the sequencing space is taken up by the dsRNA-derived reads.
We really appreciate this comment that made us reconsider our analysis. We had not previously removed small RNAs mapping to our introduced dsRNA (it was a point of debate, but we initially decided against it). We have now removed all reads targeting the respective introduced dsRNA for all samples in Figure 6 (for all sizes and from the total count) and adjusted our description and interpretation/discussion appropriately. While our overall conclusions do not change, some of the individual statistics at specific size classes did change (some were lost, possibly due to increased variance between data-sets, which affects the multiple comparisons). Out of our own curiosity and to better understand the data, we have added a new Figure panel (6d) to determine the difference between read-counts of all piRNA sized reads. While we may not see statistically significant differences at individual sizes, we can see a clear impact of most PIWI gene silencing on the sum of 24-32nt read counts. We have added this type of analysis also to Figures 7 and 8 (plus additional text) for completeness in our analysis.
Round 2
Reviewer 2 Report
I think this rivised manuscript should be accepted.